# Rechargeable potassium-ion batteries with honeycomb-layered tellurates as high voltage cathodes and fast potassium-ion conductors

Titus Masese [1], Kazuki Yoshii [1], Yoichi Yamaguchi[1], Toyoki Okumura[1], Zhen-Dong Huang[2]
Minami Kato [1], Keigo Kubota [3], Junya Furutani[4], Yuki Orikasa [4], Hiroshi Senoh [1],
Hikari Sakaebe[1] & Masahiro Shikano [1]

Rechargeable potassium-ion batteries have been gaining traction as not only promising low-cost alternatives to lithium-ion technology, but also as high-voltage energy storage systems. However, their development and sustainability are plagued by the lack of suitable electrode materials capable of allowing the reversible insertion of the large potassium ions. Here, exploration of the database for potassium-based materials has led us to discover potassium ion conducting layered honeycomb frameworks. They show the capability of reversible insertion of potassium ions at high voltages (~4 V for $K_2Ni_2TeO_6$) in stable ionic liquids based on potassium bis(trifluorosulfonyl) imide, and exhibit remarkable ionic conductivities e.g. ~0.01 mS cm$^{-1}$ at 298 K and ~40 mS cm$^{-1}$ at 573 K for $K_2Mg_2TeO_6$. In addition to enlisting fast potassium ion conductors that can be utilised as solid electrolytes, these layered honeycomb frameworks deliver the highest voltages amongst layered cathodes, becoming prime candidates for the advancement of high-energy density potassium-ion batteries.

[1] Research Institute of Electrochemical Energy (RIECEN), National Institute of Advanced Industrial Science and Technology (AIST), 1-8-31 Midorigaoka, Ikeda, Osaka 563-8577, Japan. [2] Key Laboratory for Organic Electronics and Information Displays and Institute of Advanced Materials (IAM), Nanjing University of Posts and Telecommunications (NUPT), 210023 Nanjing, China. [3] AIST-Kyoto University Chemical Energy Materials Open Innovation Laboratory (ChEM-OIL), Sakyo-ku, Kyoto 606-8501, Japan. [4] Department of Applied Chemistry, College of Life Sciences, Ritsumeikan University, 1-1-1 Noji-higashi, Kusatsu, Shiga 535-8577, Japan. Correspondence and requests for materials should be addressed to T.M. (email: titus.masese@aist.go.jp) or to Z.-D.H. (email: iamzdhuang@njupt.edu.cn) or to H.S. (email: hikari.sakaebe@aist.go.jp)

With the advent of global warming, the urgency of meeting the world's gigantic energy demands in a sustainable and ecological manner continues to grow exponentially. Lithium-ion (Li-ion) battery technology, which has been presented as an alternative solution to cater for the multifarious energy demands, is already being impeded by the threat of a projected paucity of its terrestrial mineral reserves. To address the escalating demand for large-scale rechargeable batteries, particularly for volume/weight-less-dependent applications such as the electric grid system, the development of de novo sustainable alternatives with comparable performance to that of Li-ion batteries has become a worldwide imperative[1]. Potassium (K), which is a low-cost, abundant and easily extractable element (that is, a large-ion lithophile element) unlike Li, is increasingly becoming favourable as an alternative choice. Another benefit is that K exhibits similar electrochemical properties to Li (technically speaking, $K^+$/K redox couple tends to show a lower potential than $Li^+$/Li in non-aqueous electrolytes), indicating that K-ion extraction/insertion chemistry can be applied to a similar high-voltage rechargeable battery concept as Li-ion system[2]. However, the advancement of K-ion batteries is heavily impeded by lack of materials that can facilitate the reversible insertion of the large K-ions and a limited understanding of the mechanistic aspects underlying the electrochemistry and safe utilisation of potassium-based compounds.

Potassium-based compounds are of particular interest in materials science owing to their myriad technological aspects, such as agrochemical and catalytic properties, magnetism, superconductivity, etc. Potassium adopts innumerable polyhedral coordination;[3] thus, K-based compounds have an abundance of piquant physical properties such as ferroelectricity, piezoelectricity, pyroelectricity, multiferroicity, thermoelectricity, and optics[4–13]. K-based compounds have also found their niche application in energy storage as host frameworks for reversible reinsertion of Li and Na electrodes. Such compounds include lepidocrocite $K_{0.8}Li_{0.27}Ti_{1.73}O_4$[14], $K_2Ti_6O_{13}$[15], $KFeF_3$[16], $KASO_4F$ ($A = $Fe, Co)[17], $KVPO_4F$[18], fedotovite $K_2Cu_3O(SO_4)_3$[19], $K_2[(VO)_2(HPO_4)_2(C_2O_4)]$[20], $KV_3O_8$[21], $K_{1.33}Fe_{11}O_{17}$[22], $K_2D_2(SO_4)_3$ ($D = $Cu, Fe)[19], $K_xV_2O_5$[21], Prussian analogues[23], amongst others. Indeed, a great variety of K-based minerals and compounds have been documented, and most have yet to have their electrochemical properties studied.

In spite of the largely unexplored space of K-based materials, few K-ion insertion host materials have been reported. Whilst major breakthroughs have been reported in the use of graphite[24], phosphorus[25], oxides (such as $K_2Ti_8O_{17}$[26], $K_2Ti_4O_9$[27], $Co_3O_4$-$Fe_2O_3$ composites[28], etc.) and sulphides (namely, $MoS_2$[29] and $SnS_2$[30]) as anode materials, identifying cathode frameworks that can facilitate reversible K-ion insertion is a daunting challenge. This can be attributed to the huge stress imparted in the host structures during K-ion reinsertion leading to structural collapse, as well as the difficulties in the preparation and handling of these materials due to hygroscopicity, thus debilitating the choice of cathode materials. Recently, however, feasible cathode materials such as $K_3V_2(PO_4)_3$[31], $KVPO_4F$[32], $KVOPO_4$[32,ibid] and $KVP_2O_7$[33] have been reported. They constitute the broad class of polyanionic compounds that show high voltage coupled with excellent thermal stability. A flurry of reports in recent years have also emerged on the use of organic moieties such as Prussian analogues as cathode hosts for K-ion[34–38], as they not only are of low cost, but can also deliver relatively high voltage at moderate capacities. Layered transition metal oxide analogues of classic Li- and Na-ion cathodes, for instance, $K_xMnO_2$[39,40], $K_xCoO_2$[41,42], etc. have also been reported. Amongst them, reversible K-ion insertion at moderate energy density coupled with fast K-ion diffusion rate was observed in layered metal oxides adopting P2-

type coordination (prismatic coordination of the K-ions and 2 brucite-like layers in the unit cell). P2-layered oxides with multi-transition metals such as $K_{2/3}Ni_{1/6}Co_{1/6}Mn_{2/3}O_2$[43] and $K_{2/3}Fe_{1/2}Mn_{1/2}O_2$[44] have further been shown to deliver enhanced electrochemical performance compared to their parent components (i.e., $K_xMO_2$ ($M = $Mn, Co and Fe). This information was very beneficial to us when looking for related layered cathode frameworks with greater potentials.

Pursuing new layered cathode compounds led us to investigate the growing family of honeycomb-structured multi-transition metal-layered oxides. Besides the fascinating ion-exchange aspect that leads to the synthesis of, for instance, silver-based oxides that present very fast ion mobility[45], honeycomb frameworks have been shown to be apposite structures for studying a diverse range of enigmatic magnetic properties, such as the Heisenberg–Kitaev model[46]. Therefore synthesis of new multifunctional K-based honeycomb cathode structures by suitable transition metal variations can be anticipated.

Herein, for the first time, a class of honeycomb-structured potassium-based tellurate compounds $K_{2/3}M_{2/3}Te_{1/3}O_2$ ($M = $transition metal) (hereafter denoted as $K_2M_2TeO_6$) with rich structural chemistry and interesting physical properties is being presented. Their analogues in the mineral world are the *Dagenaisite* $Zn_3TeO_6$ and *Mcalpineite* $Cu_3TeO_6$ phases that adopt honeycomb-layered frameworks[47,48]. Taking advantage of the tellurate moiety, new designs of high-voltage cathode materials can be anticipated. Moreover, the substitution of Zn or Cu with potassium (K) and $M$-3$d$ metal cations induces a transition in symmetry to hexagonal-ordered phases, which is unveiled with the structural analyses of $K_2Ni_2TeO_6$ as a prototype functional material. This study shows that these layered tellurates are not only high-voltage cathode hosts for rechargeable K-ion batteries (particularly for $K_2Ni_2TeO_6$ and its nickel-based derivatives), but are also a prolific scientific ground in the search for K-based compounds exhibiting high K-ion conductivities.

## Results

**Material characterisation and crystal structure description.** New tellurate compounds $K_2M_2TeO_6$ ($M = $Ni, Mg, Zn, Co and Cu) were synthesised by a conventional solid-state ceramics route (see details furnished in Methods section). Figure 1a presents the powder X-ray diffraction (PXRD) pattern of $K_2Ni_2TeO_6$ (equivalently written as $K_{2/3}Ni_{2/3}Te_{1/3}O_2$), for which high-quality synchrotron X-ray data were collected. Bragg peaks, taking into consideration the extinction conditions, indicate unambiguously that the titled compound crystallises in a centrosymmetric hexagonal space group ($P6_3/mcm$) with the following cell parameters: $a_{hex} = 5.2606(1)$ Å, $c_{hex} = 12.4669(3)$ Å ($V = 298.79(1)$ Å$^3$). No extra reflections from impurity phases were observed at the 0.5% detection limit. In addition, ICP-AES and SEM-EDX analysis confirmed the composition of $K_2Ni_2TeO_6$ was as anticipated (Supplementary Fig. 1 and Table 1). Inspection using high-resolution TEM (Fig. 1a inset), further reveals an array of pseudo-hexagonal symmetry dots indexable, using the hexagonal settings, as the [100] and [001] zone axes (further details are shown in Supplementary Fig. 2).

Refinements were performed assuming the crystal structure model proposed for isotypic ordered $Na_2Ni_2TeO_6$[49]. The deduced structural parameters are detailed in the Supplementary Table 2. The hexagonal crystal structure of $K_2Ni_2TeO_6$ displays layers consisting of edge-sharing $TeO_6$ and $NiO_6$ octahedra along the $ab$ planes (see Fig. 1b). Furthermore, the $NiO_6$ octahedra form regular honeycomb lattices with $TeO_6$ octahedra (located at the centre of the honeycomb lattices) as is succinctly shown in Fig. 1c. The honeycomb lattices are separated along the $c$ axis by an

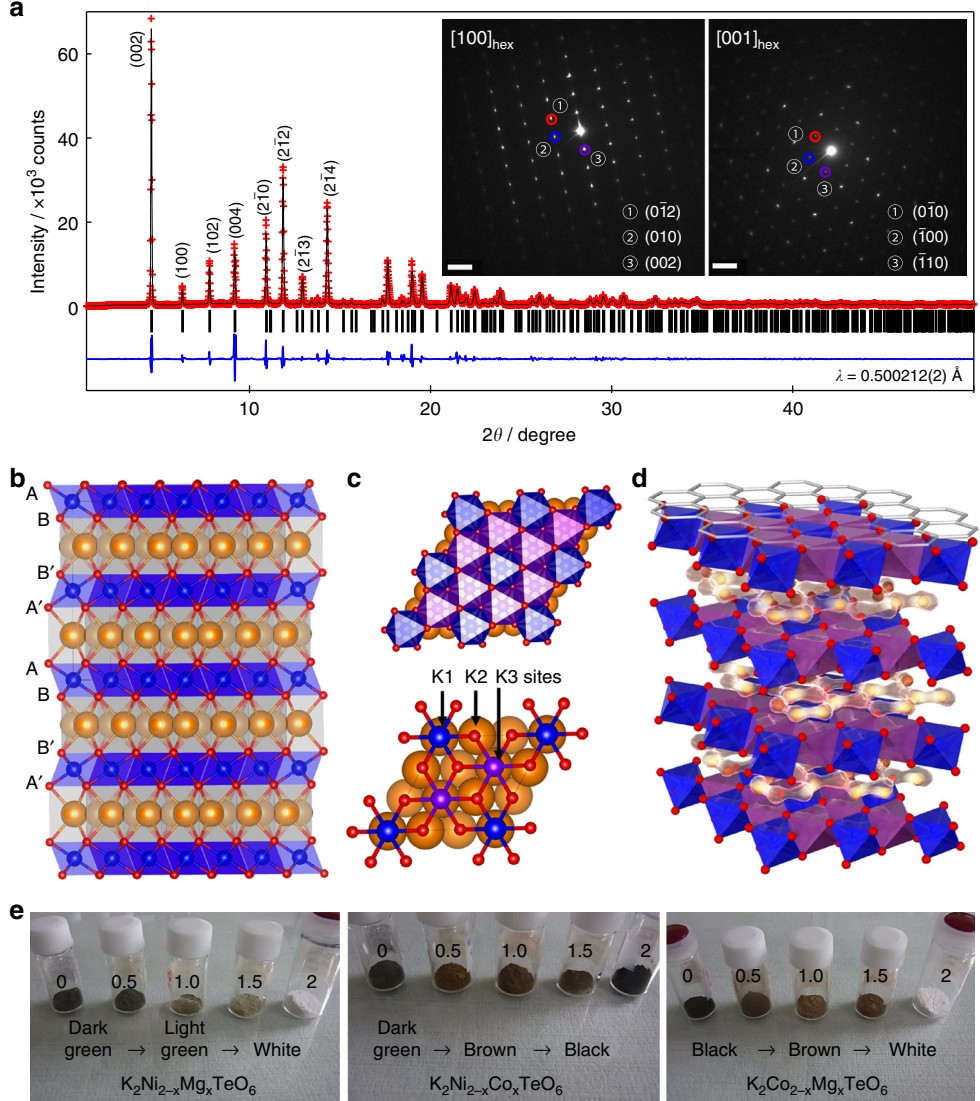

**Fig. 1** Structural characterisation of honeycomb-layered $K_2Ni_2TeO_6$ and syntheses of related solid solution derivatives. **a** Rietveld refinement data of the synchrotron X-ray diffraction powder pattern of $K_2Ni_2TeO_6$ indexed in the ordered $P6_3/mcm$ hexagonal space group ($R_B = 4.29\%$, $\chi^2 = 1.77$; low reliability values indicate a satisfactory fit). The calculated and experimental Bragg peaks are indicated in black and red, respectively. The deviation between the experimental and calculated intensity is shown in blue, whilst black ticks indicate the Bragg peaks position of the orthotellurate phase. Inset shows the corresponding SAED patterns along [100] and [001] zone axes. The scale bar shows the length scale in the reciprocal space (2 nm$^{-1}$). **b** Polyhedral view of the P2-type-layered crystal structure of $K_2Ni_2TeO_6$ along the $c$-axis: the Ni octahedra are shown in purple, Te octahedra are in blue, K ions are brown spheres and O ions are small red spheres. The prismatic polyhedra around the K ions are shown for simplicity. **c** A fragment of the $P6_3/mcm$ structure of $K_2Ni_2TeO_6$ in the $ab$ plane (the magnetoactive honeycomb layers) with three major K sites (K1, K2, and K3). **d** 2D bond valency energy landscape (BVEL) map of $K_2Ni_2TeO_6$ with isosurfaces (yellow) of 0.35 eV for K. **e** Solid solution derivatives of $K_2Ni_2TeO_6$ showing colour evolution with different $3d$ metal compositions

intermediate layer of K atoms, adopting triangular prismatic $KO_6$ coordination. This forms a P2-type stacking mode, wherein P in this nomenclature stands for trigonal prismatic coordination of the K-ions and the digit denotes the number of brucite-like layers in the unit cell.

Moreover, the present crystal structure of P2-type $K_2Ni_2TeO_6$ comprises three crystallographically distinguishable K atom sites, denoted as K1, K2 and K3, as identified in the refinement results (see also Supplementary Table 2, wherein K1, K2 and K3 wyckoff sites are designated as K3, K1 and K2, respectively, for clarity). All the three K sites adopt a prismatic coordination of oxygen atoms; three from both the bottom and top metal layers as shown in Fig. 1c. The K1 site is aligned between two face-sharing $Te^{6+}O_6$ polyhedra (each from top and bottom layers). For the K2 site, K

atoms are located between apical oxygen atoms of two $NiO_6$ and one $TeO_6$ octahedra also from both the top and bottom layers, whilst for the K3 site the K atoms reside between triangular faces of $Ni^{2+}O_6$ octahedra existing on top and bottom layers. The three K sites are only partially occupied, leading to a highly disordered distribution of K atoms residing between the honeycomb layers. A similar observation was noted recently with the analogous layered P2-type $Na_2M_2TeO_6$ phases which display superfast Na-ion conduction[49]. Bond valence energy landscape (BVEL) approach has thus been employed to garner insights into the diffusive nature of K cations within $K_2Ni_2TeO_6$. The K diffusion pathway described by the calculated isosurface of the K-ion BVEL map reveals an in-plane circular K-ion diffusion pattern within the K layers along the $ab$ plane (hence, two dimensional (2D)), as

is deduced in Fig. 1d. Note that energy of 0.35 eV above the minimum is necessary to obtain an in-plane circular 2D diffusion pattern, suggesting the feasibility of having relatively fast K-ion conduction in $K_2Ni_2TeO_6$ and related derivatives, as will be discussed in the proceeding section.

The importance of studying these layered tellurates has further been highlighted through the preparation of the complete solid solution derivatives, namely $K_2Ni_{2-x}MTeO_6$ contiguous phases ($M = Co$ and $Mg$; $x = 0$, 0.5, 1.0, 1.5 and 2.0), showing structural transitions and colour changes contingent on the combination of transition metal ions, as shown in Fig. 1e. Additional information can be found in Supplementary Figs. 3 and 4, and Supplementary Tables 3, 4 and 5. Lastly, thermal stability analyses reveal $K_2Ni_2TeO_6$ and its analogues, to be stable before reaching their melting points at above 800 °C (see Supplementary Figures 5, 6, 7 and 8), far surpassing the thermal stabilities of layered oxides such as $K_{0.3}CoO_2$[50] and polyanionic compounds such as the fluorosulphates[51]. Nevertheless, the tellurates are hygroscopic and tend to degrade upon long-time moisture exposure (see Supplementary Fig. 9). We also need to mention that the tellurate compounds detailed here show no deterioration in their physicochemical properties, under minimal exposure to moist air.

**Potassium ion dynamics and ionic conductivity measurements.** To further sense the aptitude of the tellurates towards K-ion conduction, as envisaged in $K_2Ni_2TeO_6$, diffusive barrier energies were generated based on the BVEL methodology (as aforementioned) developed by Adams and co-workers[51]. Further details are furnished in the Methods section, Supplementary Note 1 and Supplementary Fig. 10. It is important to stress that although the diffusive barrier values computed using BVEL approach have unclear physical meaning, the values are deemed legitimate in assessing the ease of cation mobility of materials relative to known compounds. The selected target materials were classified as possible cathodes and solid electrolytes, depending mainly on the constituent transition metal of compositions. A comparative plot of diffusive barrier energies of both new and documented potassium-based materials screened as shown in Fig. 2a, b, reveals very low diffusion barrier energy in the tellurate system which is comparable to $K_{0.72}In_{0.72}Sn_{0.28}O_2$ which exhibits high K-ion conductivity[52]. Further details pertaining to the K-ion diffusion pathways of the compounds are elaborated in the Supplementary Table 6 and 7. In light of this, high K-ion conductivity of the new $K_2M_2TeO_6$ tellurates in ceramic form can be anticipated, an aspect we investigated through performing ionic conductivity measurements.

In what follows, temperature-dependent alternating current (a.c.) conductivity was measured on sintered $K_2M_2TeO_6$, details of which are provided in the Methods section and Supplementary Fig. 11. Caution was exercised in performing ionic conductivity tests, considering the possible contribution from the electronic conductivity due to the presence of Ni and Co in $K_2Ni_2TeO_6$ and $K_2Co_2TeO_6$, respectively. With this caveat in mind, conductivity measurements were performed on the Mg-based analogue (i.e., $K_2Mg_2TeO_6$) which is exempt of oxidisable cation species. A plot showing the temperature dependency of the bulk conductivity (Arrhenius plot) is represented in Fig. 2c. Ionic conductivity increased with increment in temperature, as is expected, and the highest conductivity values were found at 300 °C; which marks the highest temperature we measured (see also Supplementary Fig. 11). Worthy of mention is a drastic increase in conductivity observed at around 250 °C, which has typically been noted as a phase transition onset in superionic conductors such as AgI. Further details pertaining to the phase transformation are subject of future work. The activation energy ($E_a$) calculated (between 60 and 200 °C) was about 0.92 eV (or 88.8 kJ mol$^{-1}$) as deduced upon fitting of the a.c. data with the well-known Arrhenius law and the conductivity was found to be $5.9 \times 10^{-6}$ S cm$^{-1}$ (translating to ~0.01 mS cm$^{-1}$) at room temperature (25 °C) and ~$3.8 \times 10^{-2}$ S cm$^{-1}$ (or 38 mS cm$^{-1}$) at 300 °C for $K_2Mg_2TeO_6$ (compactness of ~70%). Compared to the canonical fast K-ion conducting compounds such as $K_{0.72}In_{0.72}Sn_{0.28}O_2$ (compactness of 85%; $6 \times 10^{-7}$ S cm$^{-1}$ at 27 °C)[52], $K_2Mg_2TeO_6$ with suboptimal sample density (compactness of ~70%) demonstrates superior performance. Figure 2d shows a comparison plot of the bulk conductivities of the potassium-based tellurates with other reported K-ion superionic conductors. It is apparent that the tellurates present very high K-ion conductivity, warranting them as feasible solid electrolytes for K-ion battery. A practical contender is $K_2Mg_2TeO_6$, and further denser ceramics can, in principle, be expected to show even higher K-ion conductivities as revealed via theoretical calculations (see Supplementary Notes 2 and 3, and Supplementary Fig. 12). We are aware that the realisation of potassium-ion conductors as viable solid electrolytes for potassium-ion battery is still far from reality. However, taking into account the high reactivity and the possible dendrite formation in using alkali metal as anode materials, the development of solid electrolytes certainly will be quintessential to the commercial deployment of safe potassium-ion batteries.

**Electrochemistry of $K_2Ni_2TeO_6$ and related solid solutions/ derivatives.** The layered structure of $K_2Ni_2TeO_6$, comprising highly electronegative Ni$^{2+}$ cations in the slabs that can be oxidised during potassium extraction, calls for an evaluation of its electrochemistry. Electrochemical behaviour of $K_2Ni_2TeO_6$ and its solid solution derivatives as potential cathode materials for K-ion battery has been assessed in K half-cells using 0.5 M KTFSI in Pyr$_{13}$TFSI ionic liquid as an electrolyte, which we found to be stable during galvanostatic cycling. Further details pertaining to the ionic liquid used can be found in Supplementary Figs. 13 and 14, and the Methods section. Figure 3a displays the galvanostatic charge/discharge voltage profiles of $K_2Ni_2TeO_6$ over the first 70 cycles between 1.3 and 4.7 V at room temperature. The cells were cycled at a rate equivalent to fully charging the theoretical capacity of the material (viz. 128 mAh g$^{-1}$) in 20 h (technically written as C/20). $K_2Ni_2TeO_6$ delivers a discharge/charge capacity of ca. 70 mAh g$^{-1}$, demonstrating the ability to reversibly extract/ insert approximately one K-ion (K$^+$) ion from/into $K_2Ni_2TeO_6$, solely based on electron counting. This reversible capacity is well sustained on cycling for 70 cycles. $K_2Ni_2TeO_6$ can still sustain moderate rate performance at room temperature (shown in Supplementary Fig. 15), in spite of the inherently low ionic conductivity of the ionic liquid (0.5 M KTFSI in Pyr$_{13}$TFSI) employed. We further note that the average charge/discharge voltage is 3.6 V versus K$^+$/K; which is nearly close to the calculated voltage (see Supplementary Fig. 16). $K_2Ni_2TeO_6$ exhibits the highest voltage amongst the layered oxides reported so far (Fig. 3b). The layered $K_{2/3}Ni_{1/6}Co_{1/6}Mn_{2/3}O_2$ does not present such a high redox voltage[44]. This can be rationalised, in principle, by considering Te$^{6+}$ in $K_2Ni_2TeO_6$ as a (TeO$_6$)$^{6-}$ moiety that possesses high electronegativity than O$_2^{2-}$. The redox potential is consequently increased via the inductive effect, as has been observed for instance in polyanionic compounds such as VOSO$_4$ and VOPO$_4$ upon substitution of SO$_4^{2-}$ units by PO$_4$[53,54].

Figure 3c, d show the voltage-composition plots of $K_2Ni_2TeO_6$ solid solution derivatives, which are also electrochemically active (see also Supplementary Fig. 17), depicting high average voltages and at decent capacities. Although unclear at this point, the Mg- and Zn-substituted derivatives tend to show high reversible K-ion capacity than the Co-substituted samples, despite the high voltage

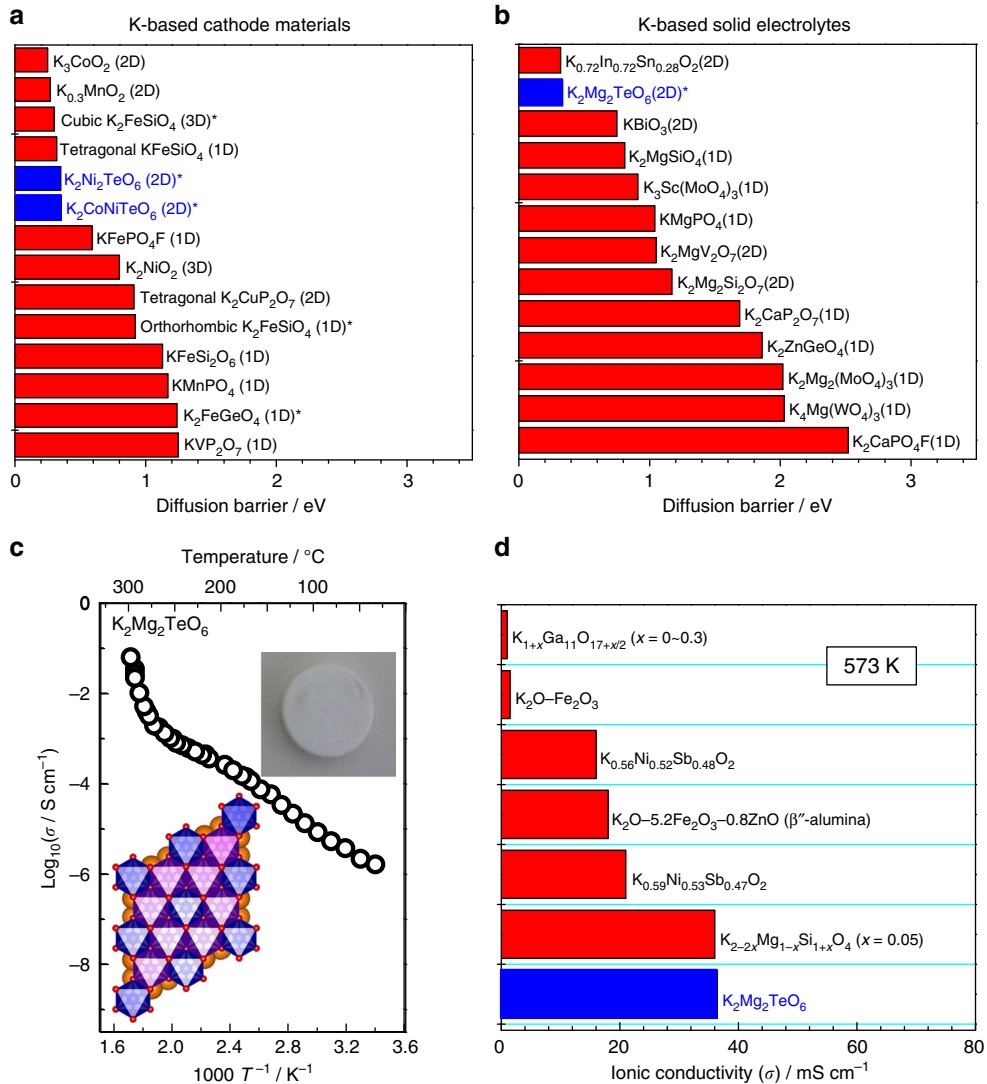

**Fig. 2** Theoretical screening of tellurate frameworks (as feasible cathodes and potassium-ion conductors) and conductivity measurements of the true ionically conducting $K_2Mg_2TeO_6$. Values of the diffusion barriers derived from BVEL isosurface maps of representative **a** potassium-based cathode materials and **b** solid electrolytes for rechargeable potassium-ion (K-ion) battery. Note that new compounds are highlighted in asterisks whereas the dimensionality of the K-ion diffusion are shown in brackets. **c** Alternating current (a.c.) bulk ionic conductivity plots for $K_2Mg_2TeO_6$ at temperature ranges of between 25 and 300 °C. **d** Bar graphs benchmarking the performance of orthotellurate compositions with the reported K-ion superionic conductors at 300 °C (573 K), references of which are provided in the Supplementary Table 8

exhibited by the latter. Figure 3e shows a plot that compares the performance of various K-ion cathode materials, based on the average working voltage, volumetric capacity and volumetric energy density attained. Cathode materials for Li-ion battery generally exhibit the highest energy density; nonetheless, several K-ion battery cathodes exhibit relatively high values that bring them to the fore as potential candidates. Layered transition metal oxides such as $K_{0.3}MnO_2$ possess compact framework and generally exhibit high reversible capacities[40], warranting them ideal for applications that require a high volumetric energy density. On another note, the inductive effect engendered by the polyanion moiety in oxyphosphates, fluorophosphates, fluorosulphates, etc., leads generally to higher voltages, as is exemplified in $KVOPO_4$[32], $KVPO_4F$[32,ibid] and $KFeSO_4F$[17]. Some organic moieties such as Prussian analogues are also promising candidates, due to the facile and fast K-ion diffusion pathways conferred by their open frameworks. However, the low density inherent in most organic moieties, encumbers their high-power utilisation when it relates to volumetric energy density. The

honeycomb-structured layered oxides, prime amongst them being $K_2Ni_2TeO_6$ and related solid solution derivatives, present a high volumetric energy density as can be envisaged from Fig. 3e, thus future research directions will be inclined in the optimisation of their performance and further exploration of related P2-type layered oxide materials.

**K-ion insertion/extraction mechanism.** The evolution of the cyclic voltammograms (namely, first and second cycle) of $K_2Ni_2TeO_6$ in K half-cells (see Supplementary Fig. 18) depicts multiple redox peaks (signature of myriad phase transitions) during K-ion insertion (extraction) into (from) $K_{2-x}Ni_2TeO_6$ coupled with $Ni^{2+(3+)}$ oxidation (reduction). For a deeper insight into the phase transition process of $K_2Ni_2TeO_6$ cathode during cycling, ex situ XRD patterns of $K_2Ni_2TeO_6$ were collected at various (dis)charging depths in K half-cells. Figure 4a–c present structural evolution of P2-type $K_2Ni_2TeO_6$ upon $K^+$ extraction and insertion. The evolution of the interslab distance, as derived

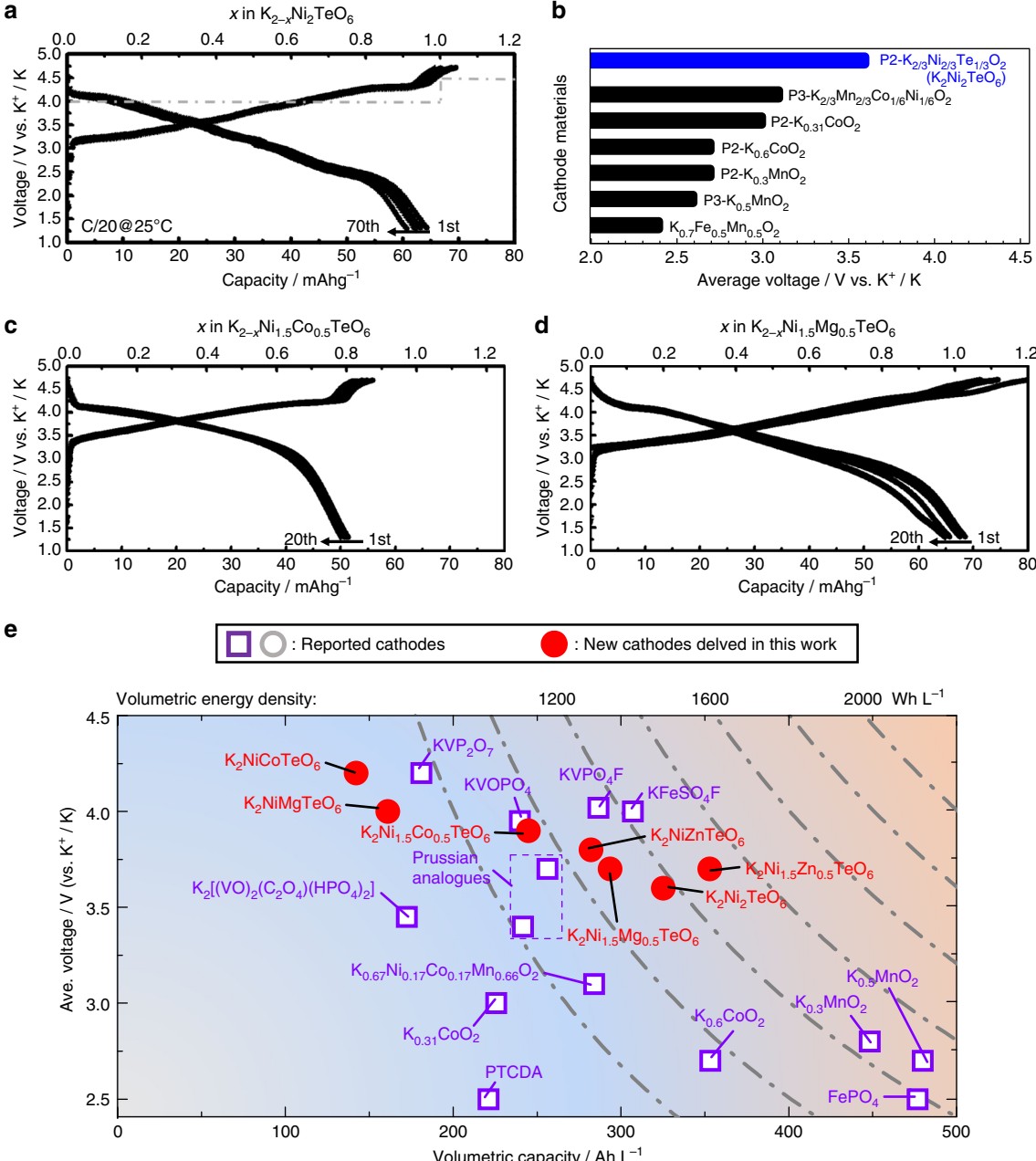

**Fig. 3** Electrochemical performance of $K_2Ni_2TeO_6$ prototype cathode material along with its $K_2Ni_{1-x}M_xTeO_6$ ($M = Mg$, Co, Zn) solid solution derivatives. **a** Voltage-capacity profiles (70 cycles) for $K_2Ni_2TeO_6$ in K half-cells using ionic liquid (0.5 M KTFSI in $Pyr_{13}TFSI$). Galvanostatic measurements were conducted at C/20 rate at room temperature. The grey dashed line indicates the calculated voltages. The 1st, 5th, 10th, 20th and 70th voltage profiles have been shown for brevity's sake. **b** Performance of new tellurate compounds benchmarked with reported layered cathode frameworks for potassium-ion battery (KIB), underlining the tellurate cathodes as high-voltage contenders. Voltage-composition plots of **c** Co-substituted $K_2Ni_{1.5}Co_{0.5}TeO_6$ and **d** Mg-substituted $K_2Ni_{1.5}Mg_{0.5}TeO_6$ derivatives. For the sake of readability, the 1st, 5th, 10th and 20th voltage profiles have been shown. **e** The average (dis) charge voltage (V versus $K^+/K$) against volumetric capacity (Ah $L^{-1}$) and volumetric energy density (expressed as Wh $L^{-1}$) for selected cathode materials for KIB. The volumetric energy densities of the cathode materials were calculated based on material density. Further details pertaining to the calculations of the selected cathode materials are provided in the Supplementary Tables 9 and 10

from full-pattern matching, is shown in Fig. 4a. (002) and (004) Bragg diffraction peaks were examined, as they are most sensitive to the $K^+$ extraction/insertion process. Analysis of other peaks aside from (00*l*), as can be noted in Fig. 4b, is difficult, owing to their low diffraction intensity. Amongst the (00*l*) Bragg peaks, shifts of the (004) peak is specifically the most discernible upon K-ion extraction (charging) and reinsertion (discharging), as observed in Fig. 4c. The (002) and (004) peaks shift to lower angles upon charging whilst the P2-type framework is retained, as

demonstrated in Fig. 4b, c. Moreover, asymmetric peak evolutions and different degrees of peak shifting are apparent, as has also been observed in P2-type $K_{0.6}CoO_2$[41] and $K_{0.5}MnO_2$[39], evincing a multitude of phase transitions accompanying K-ion extraction from $K_{2-x}Ni_2TeO_6$. At some juncture during charging, significant peak broadening occurs after which new sets of diffraction peaks appear to the left of the (002) and (004) Bragg peaks, signature of the phase transitions. Further attempts to determine explicitly the structures of the new phases hallmarking the phase transitions

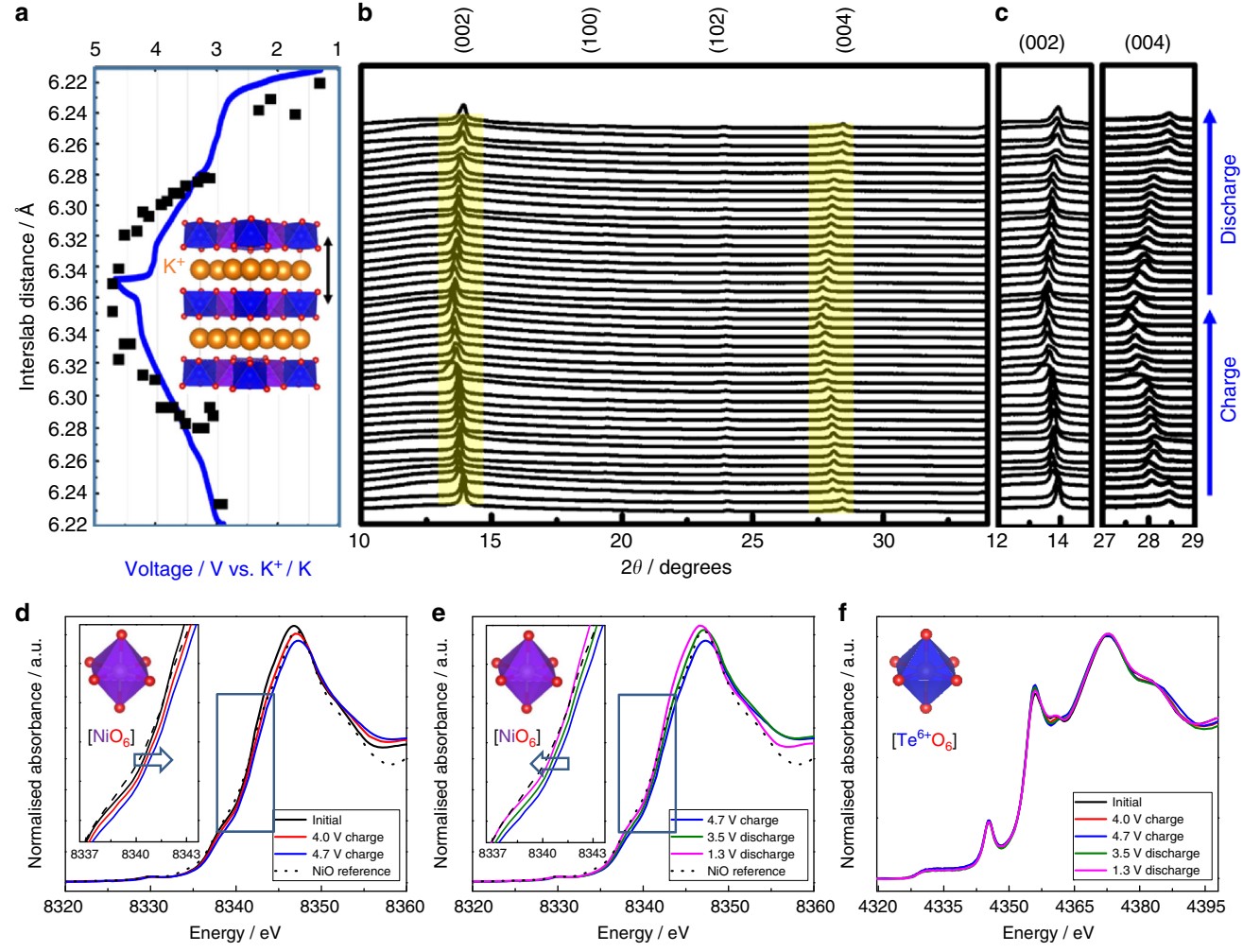

**Fig. 4** Crystal and electronic structural changes of P2-type $K_2Ni_2TeO_6$ prototype cathode material during K-ion extraction and insertion. **a** Interslab distances calculated from ex situ XRD patterns during XRD patterns whilst $K_2Ni_2TeO_6$ is (dis)charged at a current density commensurate to C/20 rate (20 h of (dis)charge). **b** XRD patterns from 10° to 38° and **c** enlarged XRD pattern indicating peak shifts of the (00l) Bragg diffraction peaks during (dis) charging. **d** Ni K-edge XANES spectra during charging (K-ion extraction) and **e** discharging (K-ion insertion). **f** Te $L_3$-edge XAS spectra of $K_2Ni_2TeO_6$ taken at different states of (dis)charging

remained elusive, apparently owing to the limitation posed by the low resolution and significant broadening of peaks. Reversal of the phase evolution is observed during the discharge process, suggesting repeatable K extraction and insertion behaviour in P2-type $K_2Ni_2TeO_6$. Noteworthy to also mention is that during the charge/discharge process, Bragg peaks tend to shift to lower/ higher angles, indicating an underlying solid solution (monophasic or single-phase) mechanism. The emergence of new peaks (as is shown in Fig. 4c) is reminiscent of a two-phase (biphasic) reaction, but these Bragg peaks also progressively shift which is typical of a monophasic reaction. A continuous increase and decrease of the c lattice parameters (viz., along [00l] direction) upon charge and discharge, respectively, is the overall trend. Upon charging (00l), Bragg diffraction peaks monotonically shift towards low angles, in principle due to the enhancement in the electrostatic repulsion between the Ni/TeO$_6$ octahedral slabs along the c-axis and vice versa during discharge. Further, a decrease/increase in the a-axis concomitantly proceeds with charging/discharging (Supplementary Table 11), due to the decrease/increase in nickel ion radius upon oxidation/reduction. The XRD results, in summary, demonstrate that P2-type $K_2Ni_2TeO_6$ can reversibly reinsert K ions through a highly reversible topotactic process, which entails not only a monophasic

but also a biphasic reaction. In addition, ex situ XRD results given in Supplementary Fig. 19 affirm that repeated $K^+$ extraction/ insertion does not affect significantly the structural integrity of the P2-type $K_2Ni_2TeO_6$ cathode material.

Apart from XRD, spectroscopic techniques were also employed to investigate the charge compensation process during reversible K-ion insertion in $K_2Ni_2TeO_6$. Generally, in order to maintain electroneutrality, the valency of the redox-active ions (in this case $Ni^{2+}$ in $K_2Ni_2TeO_6$) change upon reversible K-ion insertion. X-ray absorption spectroscopy (XAS) analysis was therefore conducted to divulge the valency changes of local chemical structure around the Ni atoms in $K_{2-x}Ni_2TeO_6$ electrodes during electrochemical cycling. As can be seen in Fig. 4d, e, the Ni K-edge XANES spectra shift to higher energy during charging and vice versa during discharging, indicating a reversible Ni valency change upon K-ion extraction and insertion. The evolution of the transition metal oxidation states of the pristine, charged, and discharged $K_2Ni_2TeO_6$ samples were confirmed further by complementary XPS analyses (Supplementary Fig. 20). Turning to tellurium, Te L-edge XANES spectra (shown in Fig. 4e) reveal no discernible spectra variations for the entire charge–discharge process, and is further confirmed by XPS studies (see Supplementary Fig. 20).

To further understand the electronic structural changes of $K_2Ni_2TeO_6$ upon K-ion extraction/insertion, theoretical calculations (more specifically, density functional theory (DFT) approaches) were carried out. Partial density of states (PDOS) of the 3d states of Ni, 4d states of Te and 2p states of O ions were calculated based on the lattice parameters of $K_2Ni_2TeO_6$. Further details of the calculations are elaborated in the Methods section and Supplementary Note 2. PDOS data for the $K_{2-x}Ni_2TeO_6$ ($x = 0$, 1 and 2) is provided in Fig. 5a. Upon K-ion extraction, $K_{2-x}Ni_2TeO_6$ tends to become metallic, implying the ease to extract K-ion from $K_{2-x}Ni_2TeO_6$ with progressive charging. The PDOS plots show the O 2p electrons to lie more in the vicinity of the Fermi energy and more itinerant than the Ni 3d electrons, whilst Te 4d bands are energetically well localised and are dormant virtually throughout the K-ion extraction process in $K_2Ni_2TeO_6$. These results suggest that, in comparison to Ni 3d, O 2p electrons are more dominant in the vicinity of the Fermi level and that the degree of hybridisation emanating from O 2p electrons in both the $K_2Ni_2TeO_6-KNi_2TeO_6$ and $KNi_2TeO_6--Ni_2TeO_6$ regimes is very large. This is conspicuous from Fig. 5a wherein, the Fermi level moves further to low-energy region upon K-ion extraction, leading to partially unoccupied up-spin bands of Ni 3d and O 2p electrons. Additionally, using the Bader scheme for topological partitioning of charge densities in the electronic structure (Fig. 5b and Supplementary Table 12), the dominance of O 2p states over the Ni 3d states and the inert role of Te 4d states in the charge transfer associated with K-ion extraction/insertion process is apparent. Overall, the PDOS plots highlight a strong hybridisation of Ni 3d and O 2p states (being dominant) during K-ion extraction process in $K_{2-x}Ni_2TeO_6$.

O K-edge XAS measurements were conducted in order to glean, experimentally, information regarding the role of oxygen during potassium-ion extraction/insertion. Although the data were acquired from ex situ measurements, caution was taken not to expose the samples to air. Normalised XANES spectra at O K-edge of $K_2Ni_2TeO_6$ during electrochemical K-ion extraction/insertion are shown in Fig. 5c. Galvanostatic measurements reveal a one-electron reaction, thus, the XAS measurements reflect the electronic structural changes occurring within the $K_2Ni_2TeO_6-KNi_2TeO_6$ regime. A clear variation of the O K-edge XAS of the $K_2Ni_2TeO_6$ cathode material before and after charging and discharging was observed. The peak around 532 eV arises from the hybridisation of O-2p and Ni-3d orbitals, whereas the peak on the high energy side emanates from the hybridisation of O-2p orbital and Ni-4s,p orbitals[55,56]. Emergence of a new peak is discernible at 528 eV (on the low-energy region) upon charging, ascribable to the formation of oxygen ligand hole. This behaviour is in excellent concordance with that observed in Li-excess

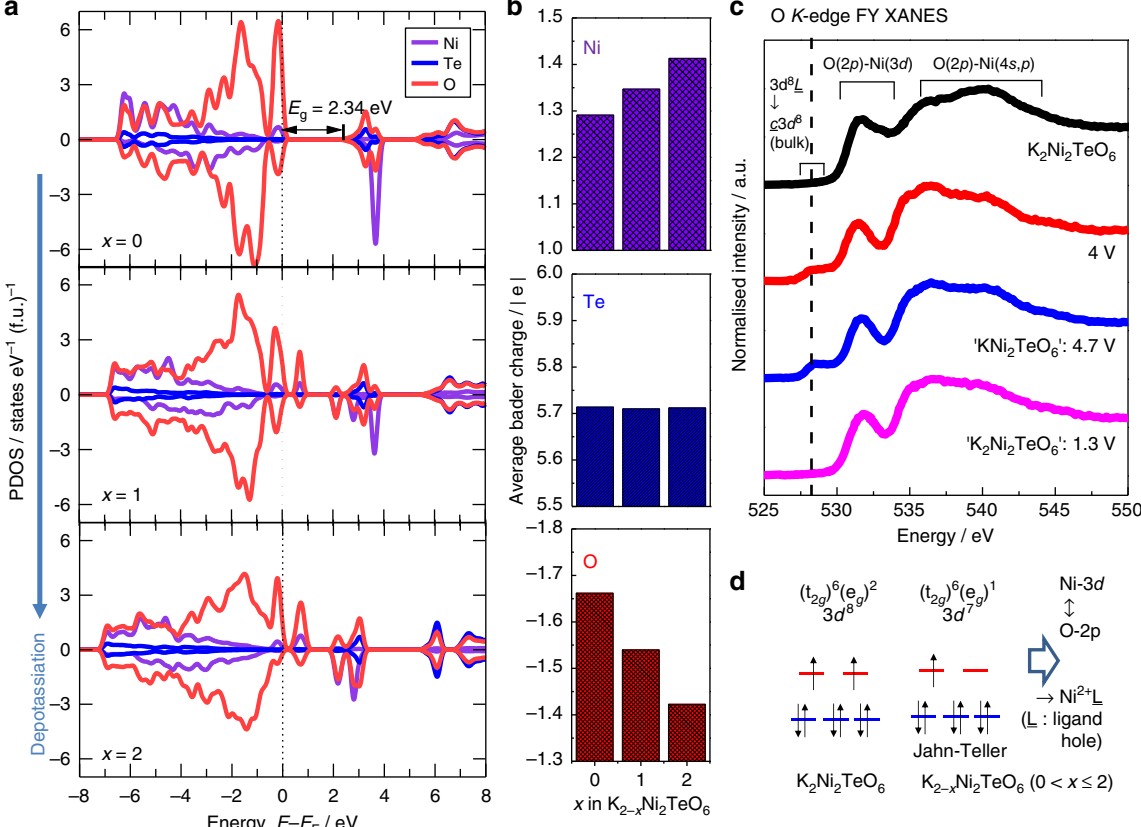

**Fig. 5** Theoretical and experimental (spectroscopic) analysis of the electronic structural changes of $K_2Ni_2TeO_6$ during K-ion extraction and insertion. **a** The computed partial density of states (PDOS) decomposed into the up-spins (positive axis) and down-spins (negative axis) for different composition $x$ in $K_{2-x}Ni_2TeO_6$ ($x = 0$, 1, 2) as function of energy ($E$). The PDOS of the up-spin nickel ($d\uparrow^5 d\downarrow^3$ at $x = 0$, $d\uparrow^{4.5} d\downarrow^3$ at $x = 1$, and $d\uparrow^4 d\downarrow^3$ at $x = 2$) and its directly bonded oxygen atoms are selected. The Fermi level (EF) is set at 0 eV with a vertical black dashed line. PDOS plots indicate a cumulative cationic (Ni 3d) and anionic (O 2p) redox activity during de-potassiation (K-ion extraction). **b** Bader charge allocation analysis on the total charge of Ni 3d, Te 4d and O 2p showing unequivocal charge fluctuations in not only Ni but also O in $K_{2-x}Ni_2TeO_6$ during K-ion extraction. **c** Normalised O K-edge ex situ XANES spectra of $K_2Ni_2TeO_6$ cathode during (dis)charging taken in fluorescence yield (FY) mode, which is highly sensitive to the innate bulk properties of $K_2Ni_2TeO_6$. Ligand holes are created in the O 2p bands during K-ion extraction and vice versa during K-ion reinsertion. **d** Schematic illustration of the electrochemistry of $K_2Ni_2TeO_6$ utilising the ligand hole formation in the hybridised Ni 3d and O 2p states during K-ion extraction

LiNiO$_2$[55], LaNiO$_3$[57] and SrNiO$_3$[58]. The peak intensity increase observed at 528 eV, in principle reflects the degree of hybridisation between Ni 3$d$–O 2$p$ orbitals, hence we can state that this strength is strongest upon K$^+$ extraction from K$_{2-x}$Ni$_2$TeO$_6$. K$^+$ extraction leads to the increase in the valency state from Ni$^{2+}$→ Ni$^{3+}$, the electronic ground state of Ni$^{3+}$ predominantly being 3$d^7$. However, owing to electronic configuration mixing triggered by the hybridisation with O-2$p$ orbitals, the ground state tends to possess a 3$d^8$ $\underline{L}$ orbital character (as has been observed in Li-excess compounds such as LiNiO$_2$[55] upon charging). Presence in the ground state of this 3$d^8$ $\underline{L}$ configuration invokes the increase in the pre-edge peak intensity conspicuous at 528 eV upon charging. It is therefore assignable to 3$d^8\underline{L} \to \underline{c}$ $d^8$ core transitions and can be rationalised, vide supra, as a signature of the hybridisation strength of O 2$p$–Ni 3$d$ orbitals[55]. In summary, the trend observed in the O $K$-edge spectra (as was a priori revealed by first-principle calculations) signifies that, hybridisation or covalency is large upon potassium extraction from K$_2$Ni$_2$TeO$_6$ and vice versa holds.

## Discussion

Potassium-ion (K-ion) batteries are a compelling grid-scale energy storage technology not only due to their low cost, but also to the attainable high volumetric energy density en par with the conventional lithium-ion batteries[59]. However, the advancement of K-ion batteries is thwarted particularly by the dearth of cathode materials that can endure repeatable K-ion (de)insertion at decent voltages and capacities. The new family of honeycomb-layered compounds with the general formula K$_2M_2$TeO$_6$ (where $M$ = Ni, Mg, Co, etc. or a combination of at least two transition metals) not only offer a fascinatingly rich playground for high-voltage cathode materials for potassium-ion (K-ion) battery (particularly for Ni-based analogues), but also enlists feasible solid electrolytes with high K-ion mobility as demonstrated in the purely ionic conducting K$_2$Mg$_2$TeO$_6$. Only few potassium-based compounds have, to date, been reported to adopt honeycomb lattice: K$_2$Mn$_2$(H$_2$O)$_2$C$_2$O$_4$(HPO$_3$)$_2$[60] and K$_2D_3$(VO$_4$)$_2$(OH)$_2$ ($D$ = Mn, Co)[61]. Orthotellurate archetypes such as K$_2M_2$TeO$_6$ augment this trove of exotic honeycomb-structured potassium-based compounds.

Besides the high volumetric capacity exhibited by P2-type K$_2$Ni$_2$TeO$_6$ and its solid solution derivatives, their record high voltages (~4 V versus K$^+$/K) in stable electrolyte (ionic liquid) and high thermal stabilities are great offsetting advantages that warrant their inclusion as high-energy-density cathode contenders for K-ion battery. Diffraction and spectroscopic techniques further reveal K$_2$Ni$_2$TeO$_6$, as a prototype cathode, can sustain successive K-ion insertion/extraction via a complex topotactic reaction. This highlights related orthotellurates as perfect K-based cathode models to study a multitude of phase transition phenomena upon re-insertion of the large K-ion.

With a theoretical capacity in the range of 120–140 mA h g$^{-1}$ and consisting of a non-terrigenous element such as tellurium (Te), it suffices to say that this study holds a more profound significance in the future application of K-ion batteries. Nevertheless, related materials can be propitious for practical applications inasmuch as a less heavy congener element to replace Te$^{6+}$ ($d^{10}$ shell occupancy) can be found. However, this study accentuates that the design of potassium-based layered cathode materials, demonstrating reversible K$^+$ (de)insertion at preponderant high voltages, is not trivial in the absence of electronegative moieties such as TeO$_6^{6-}$ and augmented with in silico material screening. The low theoretical capacities displayed by the tellurates are compensated by their high cell voltages (3.6–4.3 V versus K$^+$/K). It could be possible to assemble a high-voltage

rechargeable K-ion battery system suited for large-scale volume-restricted applications, coupling such high-voltage potassium-based cathodes to suitable anodes. A high-voltage configuration not only increases the energy density, but also reduces the number of cells per battery pack. In fact, a series connection is normally applied to smaller cells in order to achieve the required voltage of large battery packs, for instance, for those utilised in electric vehicular applications. High-voltage battery configuration results in a reduction in the number of cells, thereby curtailing the volume and cost of the battery pack. For this reason, high-voltage potassium-ion battery configurations will indeed have considerable advantage in energy storage devices.

Whilst we have shown through the preparation of solid-solution derivatives of K$_2$Ni$_2$TeO$_6$ that these high-voltage-layered honeycomb materials can be extended to include a vast variety of stoichiometries, we still feel that by heterovalent cation substitution of K$_2M_2$TeO$_6$ the highest probability exists for synthesising new honeycomb-structured potassium-based materials with superior performance. A part of an ongoing work is the design of Te-free compounds demonstrating reversible K-ion (de) insertion at high voltages and capacities. Plausible candidates encompass the potassium-rich K$_2MAO_4$, K$_3MAO_5$, K$_4MAO_6$ (or equivalently written as K$_8M_2A_2O_{12}$), K$_3M_2AO_6$ and K$_2M_2AO_6$ compounds (where $M$ and $A$ are selected from a wide variety of metals, with the summation of $M$ and $A$ oxidation states being restricted to 6, 7, 8, 9 and 10 respectively, to maintain electroneutrality). The obvious $A$ cations to try are Sb and Bi, based on the known stoichiometries of classic lithium- and sodium-based compounds. New potassium-based structural frameworks for rechargeable batteries such as the nascent K-ion battery technology still await further exploration. Moreover, it is our conviction that experimentalists and theorists looking for new research avenues in materials science, solid-state chemistry and physics will find it to be a worthwhile pursuit.

## Methods

**Synthesis**. Polycrystalline K$_2$Ni$_2$TeO$_6$ samples were prepared by solid-state reaction protocols. K$_2$CO$_3$ (Rare Metallic (Japan), 99.9%), NiO (Kojundo Chemical Laboratory (Japan), 99%) and TeO$_2$ (Aldrich, purity of ≥99.0%) were first intimately ground (using an agate mortar and pestle) in stoichiometric proportions to obtain K$_2$Ni$_2$TeO$_6$ precursor. To eliminate any adventitious water, K$_2$CO$_3$ was dried at 130 °C prior to weighing. The mixture was thereafter pelletised and fired in a gold or platinum crucible in air, argon (Ar) or oxygen (O$_2$) flow under varying duration of 6–24 h in the temperature ranges of 600–800 °C (with intermediate grinding) with a ramp rate of +200 °C h$^{-1}$. The furnace was switched off upon completion of the thermal treatment, and the samples left in the furnace to cool down. Unless otherwise stated, the obtained powders were transferred to a glove box (argon-purged), owing to the sensitivity of the materials upon exposure to air. Other related compounds of composition K$_2M_2$TeO$_6$ ($M$ = Mg, Cu, Co, Zn) were synthesised using K$_2$CO$_3$ (Rare Metallic (Japan), purity of 99.9%), MgO (Wako Chemicals (Japan), 99.9%), CuO (Kojundo Chemical Laboratory (Japan), 99%), Co$_3$O$_4$ (Kojundo Chemical Laboratory (Japan), ≥99.0%) and ZnO (Wako Chemicals (Japan), 95%) in the temperature ranges of 600–800 °C for 24–48 h. Conventional ceramic methods were also adopted to prepare the oxides with nominal compositions K$_2$Ni$_{2-x}$Mg$_x$TeO$_6$, K$_2$Ni$_{2-x}$Zn$_x$TeO$_6$, K$_2$Ni$_{2-x}$Co$_x$TeO$_6$ and K$_2$Mg$_{2-x}$Co$_x$TeO$_6$ ($x$ = 0.0, 0.5, 1.0, 1.5 and 2.0).

Powdered samples of new quaternary transition-metal-layered oxides, K$_4M$TeO$_6$ ($M$ = Co, Zn, Ni and Mg), were also confectioned by a solid-state ceramics reaction of K$_2$CO$_3$, CoC$_2$O$_4$ (Kojundo Chemical Laboratory (Japan), 99%), NiC$_2$O$_4$·2H$_2$O (Kojundo Chemical Laboratory (Japan), 99.9%), MgO, ZnO and TeO$_2$ at ~750 °C for 60 h with intermediate grinding. Synthesis of other conceivable compositions such as K$_3M_2$BiO$_6$ and K$_3M_2$SbO$_6$ were attempted under a similar firing condition by mixing MgO, CuO or ZnO and Sb$_2$O$_5$ (Aldrich, 99.999%) or Bi$_2$O$_3$ (Aldrich, purity of 99.99%) with the aforementioned precursors in stoichiometric proportions. However, only the Ni analogue of K$_3M_2$SbO$_6$ was formed as a fairly pure phase, whereas the other intermediate compounds resulted in two-phase products made up of the precursor compounds. Synthesis of high purity phases of analogous K$_2$Mn$_2$TeO$_6$, K$_2$Fe$_2$TeO$_6$, K$_2$Li$A$TeO$_6$ ($A$ = Al or Ga), K$_2$LiFeWO$_6$, K$_2M_2$WO$_6$ ($M$ = Cu or Ni), K$_2$Ni$M$SbO$_6$ ($M$ = Mn, Al or Fe) and K$_2$NiAl$T$O$_6$ (where $T$ = Ta (tantalum) or Nb (niobium)) were also in vain, whatever the synthesis route we undertook.

**X-ray diffraction analyses**. Powder X-ray diffraction (PXRD) measurements were conducted using a diffractometer (Bruker D8 ADVANCE) employing Cu–$K\alpha$ radiation (i.e., $\lambda = 1.54051$ Å). Conventional PXRD measurements were performed in Bragg–Brentano geometry with a $2\theta$ range of 10–80° at a step size of 0.01°. In addition, high-resolution synchrotron X-ray diffraction was performed at SPring-8 (Hyogo, Japan) using BL19B2 beamline with a wavelength of 0.500212(2) Å at 298 K. PXRD data analysis and refinement was carried out by the Rietveld procedure implemented in the JANA 2006 program, and the visualisation of the crystal structure was done using VESTA crystallographic software. The structural refinements for $K_2M_2TeO_6$ ($M$ = Ni, Co, Mg) were performed on PXRD data based on the $Na_2Ni_2TeO_6$ ($P6_3/mcm$, No. 194) and $Na_2Co_2TeO_6$ ($P6_322$, No. 182) structural models. Fifteen terms of Chebyshev polynomials to describe the background were used during the structural refinement and with a damping factor of 0.1. To correct peak asymmetry, Berar–Baldinozzi method and pseudo-Voigt profile function were applied. Preferred orientation with respect to the (00$l$) axis was also taken into account to obtain a satisfactory refinement. Further details regarding the crystal structure investigation(s) can be retrieved from the Fachinformationszentrum Karlsruhe, D-76344 Eggenstein-Leopoldshafen (Germany) depository database, on quoting the accession numbers CSD-434032, -434033 and -434061.

XRD ex situ measurements were collected in Bragg–Brentano geometry (Cu–$K\alpha$ radiation) for electrodes cycled at different (dis)charge depths at a rate of C/20. The trend in lattice parameters evolution during (dis)charge was determined via profile fitting of the patterns.

**Morphological and elemental characterisation**. The morphologies of the as-synthesised compounds were analysed by a scanning electron microscope (JSM-6510LA) where the elemental mapping of the constituent elements was carried out using the energy dispersive X-ray (EDX) imaging function. Transmission electron microscopy (TEM) and high-resolution images (HRTEM) were obtained on a TITAN80-300F at an accelerated voltage of 200 kV without exposure to neither air nor moisture. Simulations of the HRTEM images were performed using the JEMS 31(PECD) software. Electron diffraction experiments were conducted on many crystallites, and reproducible results were observed.

Stoichiometry quantifications and chemical compositions of the compounds were precisely determined by inductively coupled plasma absorption electron spectroscopy (ICP-AES) on a Shimadzu ICPS-8100 instrument. The measurements were repeated in several different spots of the samples, and there was no deviation in the calculated values.

**Thermal analyses**. Thermogravimetric and differential thermal analysis (TG-DTA) was carried out using a TG-DTA instrument 2020SA (Bruker AXS) in the temperature range 30–900 °C under flowing Ar or $N_2$ at a ramp rate of 5 °C min⁻¹. A baseline correction of the TG curve was carried out by measuring the empty Pt crucible prior to each measurement.

**Ionic conductivity measurements**. Pristine $K_2Mg_2TeO_6$ was uniaxially pressed into 10 mm diameter pellets under a pressure of about 40 MPa. The obtained pellets were sintered at 800 °C for 4 h in air and thereafter cooled to ambient temperature inside the kiln. This pellet densification method conserves the phase purity whilst increasing the final density of the pellet, as was confirmed by X-ray diffraction analyses on pulverised powder of the sintered pellets. The sintered pellets density was approximately 70% of the theoretical ceramics density. The pellets were coated with a thin gold film via magnetron sputtering technique. The resistance (bulk) of the sintered pellets was measured using a two-probe alternating current (a.c.) impedance spectroscopy (Solartron SI 1260) over a frequency ranging from 0.01 Hz to 1 MHz at a perturbation of 100 mV. The impedance analyser was connected to a purpose-built apparatus that facilitated measurements to be carried out at various temperature ranges under argon. Impedance spectroscopic data were collected between 25 and 300 °C with impedance scans taken every 10 °C. Ionic conductivities (of the bulk) recorded at various temperatures were obtained by Nyquist plot fittings. Arrhenius graphs were plotted taking into consideration the errors arising from area measurements, device inductance (that was observed at high temperatures), and the impedance frequency analyser. The activation energy ($E_a$) for K-ion conduction was calculated through a linear fitting of the bulk ionic conductivity values at various temperatures by incorporating the well-established Arrhenius equation: $\sigma = \sigma_0 \exp(-E_a/k_BT)$, whereby $E_a$ represents the activation energy (in this case, for K-ion conduction), $\sigma_0$ as the absolute ionic conductivity (at zero temperature), $\sigma$ denotes the temperature-contingent ionic conductivity, whilst $T$ and $k_B$ have their conventional definitions.

**Theoretical calculations**. BVEL calculations were conducted using the bond valency parameters developed by Adams and co-workers[51], which augment the polarisability of the mobile species, in this case $K^+$, and the influence of the counter ions of the structure up to a distance of 10 Å. This approach allows the visualisation of conduction trajectories within the structure whilst giving hints to possible K-ion conduction mechanisms.

DFT calculations were conducted based on the generalised gradient approximation (GGA) with the Perdew–Burke–Ernzerhof (PBE) exchange-correlation functional and projector augmented wave (PAW) pseudopotentials as implemented in the Quantum-ESPRESSO (QE) Package. A plane-wave cutoff of 60 Rydberg (Ry) for the kinetic energy and 500 Ry for the charge density were used to expand the wave functions. Full cell relaxation for all the present structures were conducted with a tolerance of $10^{-10}$ Ry for total energy and $10^{-3}$ eV Å⁻¹ for the residual forces on each ion. Spin-polarised GGA + $U$ approach was applied for $K_{2-x}Ni_2TeO_6$ ($x$ = 0, 1, 2) with an antiferromagnetic ordering of Ni akin to that of $Na_2Ni_2TeO_6$ (adopting the same $P6_3/mcm$ symmetry); where the effective Hubbard parameter $U_{eff}$ ($= U - J$) for Ni $d$ orbitals were taken to be 6 eV[62]. Further details regarding the theoretical approach implemented can be found in the Supplementary Notes 1 and 2.

**Electrochemical measurements**. Assembly of the coin cells and pertinent protocols were carried out inside an Ar-filled glove box (MIWA, MDB-1KP-0 type) with $H_2O$ and $O_2$ contents <0.1 ppm. Pertaining to electrode fabrication, $K_2Ni_2TeO_6$ and related tellurates were mixed with polyvinylidene fluoride (PVdF) binder and carbon to attain a final weight formulation of cathode material: binder: carbon as 85: 7.5: 7.5. A viscous slurry was made by suspending the mixture in $N$-methyl-2-pyrrolidinone (NMP), which was then cast on aluminium foil with a mass loading of 4–5 mg cm⁻². Composite cathodes, with a geometric area of 1 cm², were punched out and dried in vacuo at 120 °C. Electrochemical performances were assessed in CR2032-type coin cells using $K_2Ni_2TeO_6$ composite cathode (and related tellurates), separated from the K metal anode (counter electrode in technical terms) by glass fibres discs soaked with electrolyte. Owing to the pronounced decomposition of the common electrolytes like 0.8 mol dm⁻³ $KPF_6$ (Potassium hexafluorophosphate) (Sigma Aldrich, 99.99%) in propylene carbonate (PC)/ethylene carbonate (EC) (1:1 volume %), diethylcarbonate (DEC)/EC (1:1 volume %) or 1 mol dm⁻³ KFSI (potassium bis(fluorosulfonyl)imide) in dimethylcarbonate (DMC)/EC (1:1) that we experimented, we have used a 0.5 mol dm⁻³ potassium bis (trifluoromethanesulfyonyl)imide (hereafter denoted as KTFSI) in 1-methyl-1-propylpyrrolidinium bis(trifluoromethanesulfonyl)imide (hereafter Pyr₁₃TFSI) (Kanto Chemicals (Japan), 99.9%, <20 ppm $H_2O$) ionic liquid, which was more stable against decomposition at high voltages and compatible with most potassium-based cathode materials. Unless stated otherwise, galvanostatic cycling was done at a current rate corresponding to C/20 (20 being the necessary hours to extract/insert 2 $K^+$ per formula unit), and cyclic voltammetry tests were conducted between 2.8 V and 4.6 V (versus $K^+$/K) and at a scan rate of 0.1 mV s⁻¹. All electrochemical measurements were performed at room temperature.

**X-ray photoelectron spectroscopy (XPS)**. XPS analysis was performed on a PHI 5000 Versa Probe (Ulvac-Phi) instrument. The hygroscopic cathode samples were loaded onto a spectrometer and transferred into the chamber in a moisture-free environment. Gaussian functions were used to fit all the XPS spectra and data processing was performed using COMPRO software. The binding energy values of the obtained spectra were calibrated, as is customary, based on the C 1$s$ peak at 285 eV corresponding to hydrocarbon. XPS spectra of pristine, charged (4.0 and 4.6 V) and discharged electrode samples (to 1.3 V) were collected at Ni 2$p$ and Te 3$d$ binding energy.

**X-ray absorption spectroscopy (XAS)**. For X-ray absorption measurements, ex situ measurements were also performed because high-quality oscillated data were required. Reference compounds were prepared as homogeneous self-supporting pellets. For the Ni $K$-edge and Te $L_3$-edge XAS measurements, charged/discharged $K_{2-x}Ni_2TeO_6$ electrodes were hermetically sealed in packets inside a glove box (Ar-purged). The XAS spectra were measured in the Ni $K$-edge and Te $L_3$-edge energy region (at room temperature) in transmission mode at BL01B1 of SPring-8 (Japan) synchrotron facility. Athena package was used to treat the raw X-ray absorption data, as is customary. As for the O $K$-edge and Ni $L$-edge XAS measurements, the (dis)charged electrode samples were transferred to a measurement vacuum chamber without air exposure. The spectra were measured in fluorescence yield mode (which is sensitive to the bulk state of a sample) using the beamline facility (BL-2) of SR Center located at Ritsumeikan University (Japan).

## Data availability

Data that support the findings detailed in this study are available in the supplementary information and its article. Any other source data perceived as pertinent are available, on reasonable request, from the corresponding author [T.M.].

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

## Acknowledgements

We gratefully acknowledge Dr. Godwill Mbiti Kanyolo, Dr. Stephen Njane, Dr. Linus Atisa, Mr. Tuesday Masaki, Ms. Josephine Mokeira, Ms. Catherine Nyaboke and Dr. Sou Taminato for useful discussion, particularly in giving an 'outsiders' perspective of the paper. We thank Dr. Keiichi Osaka for conducting synchrotron X-ray diffraction measurements at SPring-8 facility (Proposal number 2017B1773). Dr. Mineyuki Hattori, Dr. Shigenobu Hayashi and Dr. Yoshito Gotoh are thanked for help in conducting [125]Te NMR measurements. We also acknowledge Ms. Kumi Shiokawa and Ms. Yumi Haiduka for advice and technical help during conducting the electrochemical, XRD and XAS measurements. Mr. Yoshinari Sakaguchi, Ms. Kimiko Sakaguchi and Ms. Natsumi Ishii are also thanked for fruitful comments.

## Author contributions

T.M. contributed to the syntheses of most of the potassium-based compounds. K.K. performed the morphological and thermal characterisation of the compounds. Z.-D.H. and T.M. contributed to the crystal structural analyses of the materials. Y.Y performed the DFT calculations. T.O. performed conductivity measurements. K.Y. prepared the ionic liquids. K.Y., M.K. and T.M. performed the electrochemical measurements. K.Y., H.Sak. and H.Sen. interpreted the electrochemical measurement results. T.O. and T.M. performed XRD and hard XAS measurements of the electrodes. Y.O. and J.F. performed soft XAS measurements and interpreted the results. M.S. and K.Y. conducted XPS measurement of the electrodes. T.M., Y.O., Z.-D.H and H. Sak. wrote the manuscript. M.K. designed most of the graphics outlined in the work. H.Sen., H. Sak. and M.S. provided input with the data analysis, helped with the discussion and assisted with the manuscript correction. The ideas and experiments were conceived, analysed and planned by all co-authors under the supervision of M.S. All authors have given approval to the final version of the manuscript.

## Additional information

**Competing interests:** The authors declare no competing interests.

