## [Peer Review file · Nature Communications]

Reviewers' comments:

Reviewer #1 (Remarks to the Author):

The authors, for the first time, reported the new potassium-based layered honeycomb frameworks ($K_2M_2TeO_6$, where M = transition metal). The as-prepared $K_2M_2TeO_6$ not only show reversible K-ion insertion at high voltages (namely, ~ 4 V versus K-ion for $K_2Ni_2TeO_6$) in stable electrolyte based on potassium bis(trifluorosulfonyl) imide ionic liquid, but also exhibit relatively fast K-ion conductivities (~ 0.01 mS cm⁻¹ (at 298 K) and ~ 40 mS cm⁻¹ (at 573 K) for $K_2Mg_2TeO_6$) by far surpassing most of known K-ion superionic conductors reported to date. This work provides new layered honeycomb-structured frameworks that deliver the highest voltages on record of layered cathode frameworks hitherto reported for KIBs, in addition to enlisting a new class of fast K-ion conductors that can be utilised as solid electrolytes towards the advancement of high energy density and safe potassium-ion batteries.

It can be published in Nature communications after addressing the following questions:

- 1) . Overall, the organization of the manuscript could be improved. The early part of the manuscript discusses the structural aspects of the materials, which does not necessarily help the readers to understand the mechanistic aspect of the electrode in this material.
- 2). For the part on electronic structure, my main technical concerns are detailed respect to the justification of the methods used.
- 3). Pls. provide the crystal structure in Fig. 1 or merge the Fig. 1 and Fig.2 to demonstrate the atoms arrangement for benefiting the readers understand the symmetry of the as-prepared materials.
- 4). Can the authors confirm the index of SAED (Fig.1 c and SI Fig. 3). If the materials have the P63/mcm hexagonal space group, it shouldn't show hexagonal patterns along the [010] zone axis.
- 5). The authors claimed that: " A redox peak, presumably relating to oxidation of Ni²⁺ to Ni³⁺, appears at around 4.3 V. The appearance of this redox peak and humps was also observed during the discharge process showing the reversibility of the electrochemical process. The redox peaks are broad, suggesting a single-phase mechanism upon cycling. Single-phase mechanism in this context means that only lattice volume contraction or expansion occurs, with no remarkable global structural change." However, according to the CV curve of the Fig. 4b, at least another 3 redox peaks can be observed. Therefore, it is hard to believe the as-prepared materials go through the single-phase mechanism.
- 6). The authors gave the ex-situ XRD of the specific charge and discharge states at 4, 4.7, 3, and 1.3 V. But it is not sufficient to verify the crystal change between these particular voltages. Therefore, it is recommend to collect the in-situ XRD instead and help to identify the phase change mechanism during the charge and discharge processes.

Reviewer #2 (Remarks to the Author):

There has been a wave of interest recently in "honeycomb" variants of the layered oxide structures that dominate Li-ion and Na-ion battery cathode applications. The paper under review expands the scope of that interest to K-ion batteries, another growth area in solid-state electrochemical research. While it is arguable whether K-ion batteries have genuine commercial potential, given the obvious challenges such as slow kinetics and extreme hydrophilicity, I have no doubt that this work will attract attention from the huge number of battery researchers looking for new avenues to explore.

While the compounds described are certainly interesting, some of the claims about their properties are hyperbolic, or at least, framed very optimistically. High voltages and the resulting need for ionic liquid electrolytes entails as many disadvantages as advantages. It is difficult to imagine $K_2Mg_2TeO_6$ really

being used as a solid electrolyte, given the difficulties in making solid electrolytes commercially viable in the more favourable Li and Na cases. One particular claim that struck me was that "their high densities ... improves the volumetric energy density" - which is misleading because the high density is due to the presence of a heavy metal (Te), not to a more closely-packed structure. The performance of these materials should be described realistically - they are undeniably good K conductors, their thermal and cycling stabilities are excellent, they could theoretically perform better, and (perhaps most importantly) they open up new chemistry to exploration.

Technically, the work is very thorough and well done. No additional experiments are needed to convince me that the key new compounds synthesised and tested ($K_2Ni_2TeO_6$ and $K_2Mg_2TeO_6$) have been correctly described in terms of their chemical, crystal and electronic structures. Additional compounds are described in sufficient detail in some cases (e.g., the Ni/Mg/Co solid-solutions), but obliquely in others. For example, the Discussion ends with a mention of other "plausible candidates" including $K_3M_2AO_6$ and $K_2M_2AO_6$, without explicitly suggesting what A might be. The obvious A cations to try are Sb and Bi, based on the known Na (for K) analogues; and, indeed, the Synthesis section mentions that $K_3M_2BiO_6$ and $K_3M_2SbO_6$ were prepared. However, the former is not mentioned anywhere else in the paper (as far as I can tell) and the latter (as $K_3Ni_2SbO_6$) only appears in the SI, with no useful additional commentary. This suggests a desire to hold a significant amount of work back for later publication, while still laying claim to the first synthesis here. While I recognise that the paper is already quite long, I feel that some more transparency concerning these additional compounds would be desirable.

In summary, the work is interesting and of excellent quality. The technical content is publishable in its present form in a good journal. Its suitability for an "all of science" journal is somewhat marginal, and this is reflected in the need to describe the work in a very optimistic way, rather than let the results speak for themselves. I would not be upset to see this published in Nature Communications, but err on the side of it being more suitable for a slightly more specialised (top-tier) materials or energy focused journal.

Reviewer #3 (Remarks to the Author):

This manuscript reports the synthesis and performance tests of a group of tellurate compounds toward their possible application in K-ion batteries. K-ion batteries is an interesting topic and good cathode materials may attract attention of the field. The experimental works are in general quite solid, despite a few minor flaws. Two overarching concerns need to be addressed, besides detailed technical questions listed below. 1. The main focus of this paper seems to be blur with both the ionic conductor material and cathode material presented. It would be better if only one functionality/application is presented with deep discussion than two materials being presented with general discussion. The ionic conductor/cathode can be separated as two papers. Also, it would be totally worth it to publish the information in the supplementary information as papers in the future. These are valuable information but nowadays few people get to read the supplementary info. 2. Tellurium is a very expensive element, which implies the tellurates may never be really commercialized for K-ion batteries. The reviewer totally understands fundamental and exploration type of research should not be limited by application considerations. However, more discussion on the general impact of this work in K-ion battery and energy storage is necessary for a broad readership. Detailed comments/questions are as follows:

1. The reviewer strongly suggest the authors to follow the guide for authors to write the manuscript. For example, this current manuscript does not have page number, making the reviewing painful.
2. Page 3. Typo: $K_2/3M_1/3Te_1/3O_2$ should be $K_2/3M_2/3Te_1/3O_2$

3. A completely in-plane ordering of Ni and Te was used in the refinement with putting Ni and Te at different sites. Did the author check with refinements that puts Ni and Te in random distribution in P63mcm space group to see if it is possible? The ionic radius of Ni²⁺ is 0.69 Å and Te⁶⁺ is 0.56 Å in octahedral sites. I guess they would prefer to order from both size and charge coupling considerations, but some degree of disordering still may exist, which could give significant impact to the diffusion barriers. The disordering of K⁺ at K1 K2 and K3 sites seems also indicating the disordering in the Ni-Te plane. Local structure sensitive characterizations such as Pair Distribution Function or solid state NMR may be considered to complement XRD refinement.
4. In addition, it would be great if the authors can provide the cif files of the crystal structure after the refinements, as part of the supplementary information.
5. Figure 1a is too small to read. A figure of refinement could be very informative but not with this small size. Suggest to either increase the size in main text, or give a high resolution figure in supplementary info.
6. How hygroscopic are the tellurates? How long has the sample been exposed to air before the XRD in figure S11 was collected? Minutes, hours, or days?
7. The computed in-plane diffusive energy barrier is 0.35 eV, while the experimental measured activation energy is 0.92 eV. Although they are not exactly the same energies/barriers, they commonly do not deviate that much in other ionic conductors. Could the authors explain this?
8. How is the long term stability of the cathode material? Could the author provide an ex situ XRD of the material after some cycles?
9. The EIS spectra of K₂Mg₂Te₂O₆ in 25°C and 100°C are expected to show semi-circles with larger radii than those of 200°C and 300°C. Why do they only show dots in very low impedance area?
10. It is suggested that the spectra of the standards, such as NiO, should be also shown in the Ni XANES results (figure 6c) for reader to gauge the increase of the valence of Ni.
11. The space in Figure 6b is not well used to better demonstrate the details as 6a and 6b are not that different. It would be much better to show in 6b only zoom-in of two peaks with the change in their 2θ.
12. The authors claimed that the reversible capacity is partly due to O-redox based on the O-K-edge XANES. This is a weak argument. Since the O 2p and Ni 3d orbitals are hybrid, it is arguable to say this is an O-redox. Similar arguments have been extensively discussed in Li-excess cathode materials in Li-ion batteries.

COMMENTS FROM REFEREE #1:

The authors, for the first time, reported the new potassium-based layered honeycomb frameworks ($K_2M_2TeO_6$, where M = transition metal). The as-prepared $K_2M_2TeO_6$ not only show reversible K-ion insertion at high voltages (namely, ~ 4 V versus K-ion for $K_2Ni_2TeO_6$) in stable electrolyte based on potassium bis (trifluorosulfonyl) imide ionic liquid, but also exhibit relatively fast K-ion conductivities (~ 0.01 mS cm^{-1} (at 298 K) and ~ 40 mS cm^{-1} (at 573 K) for $K_2Mg_2TeO_6$) by far surpassing most of known K-ion superionic conductors reported to date. This work provides new layered honeycomb-structured frameworks that deliver the highest voltages on record of layered cathode frameworks hitherto reported for KIBs, in addition to enlisting a new class of fast K-ion conductors that can be utilised as solid electrolytes towards the advancement of high energy density and safe potassium-ion batteries.

Thank you for your thoughtful and thorough review of our manuscript. Benefitting from the revisions suggested, the revised manuscript now has new information and additional interpretations.

Major comments:

It can be published in *Nature Communications* after addressing the following questions:

Thank you again for your encouraging comments

1) Overall, the organization of the manuscript could be improved. The early part of the manuscript discusses the structural aspects of the materials, which does not necessarily help the readers to understand the mechanistic aspect of the electrode in this material.

Response: We appreciate your comments greatly. To address the concern raised, we have revised Figure 1 as you will find in the revised manuscript (page 5). **Figure 2 has been merged into Figure 1** to succinctly demonstrate the atoms arrangement, which is rudimental to understanding the mechanistic aspects of this material, for instance, the volume expansion and contraction occurring during reversible potassium re-insertion and the nature of potassium ion trajectory. In addition, we have rewritten the results to elucidate the link between structural aspects of $K_2Ni_2TeO_6$ and the mechanistic aspects demonstrated upon K-ion extraction / insertion, as readily divulged by XRD measurements.

2) For the part on electronic structure, my main technical concerns are detailed respect to the justification of the methods used.

Response: Thank you for giving us an opportunity to allay concerns pertaining to the justification of the methodologies used to analyse the electronic structural changes. We have employed both **hard and soft X-ray techniques** to elucidate the electronic

structural changes in the prototype $\text{K}_2\text{Ni}_2\text{TeO}_6$ during successive K^+ extraction and insertion. Particularly, **Ni K-edge XAS** (hard X-rays) has been used to probe the valency changes in $\text{K}_{2-x}\text{Ni}_2\text{TeO}_6$. The XAS results have been complemented by XPS (although amenable to the surface structures). Such an approach has been well used. For instance, Tarascon and co-workers used hard X-rays corroborated with XPS to analyse the electronic structural changes in $\text{Li}_2\text{Ru}_{1-y}\text{Sn}_y\text{O}_3$ and $\text{Li}_{4-x}\text{NiTeO}_6$ upon (dis)charge¹⁻³.

The role of Te to the charge compensation process has been probed by **Te L-edge XAS** as well as XPS, revealing univocally Te^{6+} as a spectator ion. Te *L*-edge has been shown to be a potent spectroscopic tool in accessing the change in valency of Te in compounds, for instance in reference 4. To probe the role of oxygen to the charge compensation process, soft X-rays (*viz.*, **O K-edge XAS**) taken in the bulk-sensitive **fluorescence yield (FY) mode** were employed. This approach has been used, for example, in references 5 and 6. We have also detailed the methods used in this study in the METHODS section of the main manuscript (page 25-line 865). Additionally, well-established **theoretical approaches** have been utilised to further glean *a priori* on the charge compensation processes in the titled compound upon repeated K^+ (de)insertion. The efficacies of the theoretical approaches used in the analysis of the electronic structures have well been documented. The experimental protocols are detailed in the METHODS section (revised manuscript page 23) and Supplementary Information (Supplementary Notes 2 and 3). We have also provided new spectroscopic analyses data in the RESULTS section (page 14, Figures 4d-4f)

Reference

1. Tarascon, J. -M. *et int.* $\text{Li}_4\text{NiTeO}_6$ as a positive electrode for Li-ion batteries. *Chem. Commun.* **49**, 11376 (2013).
2. Tarascon, J. -M. *et int.* Reversible anionic redox chemistry in high-capacity layered-oxide electrodes. *Nat. Mater.* **12**, 827-835 (2013).
3. Tarascon, J. -M. *et int.* Fundamental interplay between anionic/cationic redox governing the kinetics and thermodynamics of lithium-rich cathodes. *Nat. Commun.* **8**, 2219 (2017).

4. Khanna, A. *et al.* Structural analysis of WO_3 - TeO_2 glasses by neutron, high energy X-ray diffraction, reverse Monte Carlo simulations and XANES. *J. Non-Crystalline Solids* **495**, 1016 (2018).
5. Yabuuchi, N. *et al.* Origin of stabilization and destabilization in solid-state redox reaction of oxide ions for lithium-ion batteries. *Nat. Commun.* **7**, 13814 (2016).
6. Yoon, W. -S. *et al.* In situ soft XAS study on nickel-based layered cathode material at elevated temperatures: A novel approach to study thermal stability. *Nat. Commun.* **4**, 6827 (2014).

3) Pls. provide the crystal structure in Fig. 1 or merge the Fig. 1 and Fig.2 to demonstrate the atoms arrangement for benefiting the readers understand the symmetry of the as-prepared materials.

Response: The merging of the figures has indeed helped shorten the manuscript. Thank you greatly for your suggestion. As recommended by the author, **Figure 2 has now been merged into Figure 1** to explicitly demonstrate the atoms arrangement thus benefiting the reader's understanding of the symmetry of the as-prepared $\text{K}_2\text{Ni}_2\text{TeO}_6$.

4) Can the authors confirm the index of SAED (Fig.1 c and SI Fig. 3). If the materials have the $P6_3/mcm$ hexagonal space group, it shouldn't show hexagonal patterns along the [010] zone axis.

Response: Without a doubt, the reviewer is right in that the electron diffraction patterns should not show hexagonal patterns along the [010] zone axis. **It ought to be the [001] zone axis**, as shown below. We have corrected this in the revised manuscript. We apologise for the error inadvertently made in labelling the zone axis.

Figure 1. Electron diffraction patterns recorded on (a) $\text{K}_2\text{Ni}_2\text{TeO}_6$ indexed in a hexagonal lattice along $[100]$ and $[001]$ directions.

These results have been added to RESULTS section and Supplementary Information section of our revised manuscript (manuscript page 6-Figure 1 and Supplementary Figure 2).

5) The authors claimed that:” A redox peak, presumably relating to oxidation of Ni^{2+} to Ni^{3+} , appears at around 4.3 V. The appearance of this redox peak and humps was also observed during the discharge process showing the reversibility of the electrochemical process. The redox peaks are broad, suggesting a single-phase mechanism upon cycling. Single-phase mechanism in this context means that only lattice volume contraction or expansion occurs, with no remarkable global structural change.” However, according to the CV curve of the Fig. 4b, at least another 3 redox peaks can be observed. Therefore, it is hard to believe the

as-prepared materials go through the single-phase mechanism.

Response: Thank you greatly for the suggestion. We concur with the reviewer that it is not convincing to the readers that $K_2Ni_2TeO_6$ undergoes a single-phase mechanism, whereas multiple redox peaks are discernible in the cyclic voltammogram. We have deleted this sentence since additional XRD measurements we performed now reveal a complex process entailing a combination of biphasic and monophasic (solid solution) behaviour during galvanostatic cycling of $K_2Ni_2TeO_6$. This mechanism was missing in the former version of the manuscript, and we feel that it should be definitively included in the new version of the manuscript. Please see the XRD measurements detailed in comment (6).

“[...] A redox peak, presumably relating to oxidation of Ni^{2+} to Ni^{3+} , appears at around 4.3 V. The appearance of this redox peak and humps was also observed during the discharge process showing the reversibility of the electrochemical process. The redox peaks are broad, suggesting a single-phase mechanism upon cycling. Single-phase mechanism in this context means that only lattice volume contraction or expansion occurs, with no remarkable global structural change.”,

This sentence has been deleted. It now reads as follows:

“The evolution of the cyclic voltammograms (namely, first and second cycle) of $K_2Ni_2TeO_6$ in K half-cells (see Supplementary Figure 18) depicts multiple redox peaks, which neatly superimpose revealing the reversibility of K-ion insertion (extraction) into (from) $K_{2-x}Ni_2TeO_6$.coupled with $Ni^{2+(3+)}$ oxidation (reduction).”

This sentence has been added to the RESULTS section of our revised manuscript (page 13, line 443-line 446).

6) The authors gave the *ex-situ* XRD of the specific charge and discharge states at 4, 4.7, 3, and 1.3 V. But it is not sufficient to verify the crystal change between these particular voltages. Therefore, it is recommend to collect the *in-situ* XRD instead and help to identify the phase change mechanism during the charge and

discharge processes.

Response: The reviewer has raised an important aspect that is worth investigating. To get a more holistic picture of the multitude of phase transitions, as surmised from the cyclic voltammograms, we have performed additional XRD experiments during charge and discharge of $\text{K}_2\text{Ni}_2\text{TeO}_6$.

Taking caution on the potential safety risk associated with the high reactivity of potassium metal, designing not only a safe *operando* cell but that which can reliably avail fairly high quality *in situ* XRD data proved elusive. Complementary information about the phase evolution during cycling was therefore obtained from *ex situ* X-ray diffraction studies at a selection of various points collected at almost every 0.1 V voltage change during the charge and discharge processes. The measurement protocols and results have been added to RESULTS and METHODS section of our revised manuscript (page 14, revised Figure 4a and page 22, line 746).

Figure 3. (a) Typical voltage (dis)charge profiles of P2-type $\text{K}_2\text{Ni}_2\text{TeO}_6$ at a current density commensurate to C/20. Interlayer distances calculated from XRD patterns during (dis)charge processes are also shown. (b) XRD patterns from 10° to 38° . (c) Enlarged images of (00 l) Bragg peaks obtained from XRD.

These results have been added to RESULTS section of our revised manuscript (page 14, line 451-page 15, line 515 and revised Figure 4a).

“Figures 4a, 4b and 4c present structural evolution of P2-type $K_2Ni_2TeO_6$ upon K^+ extraction and insertion. The evolution of the interslab distance, as derived from full-pattern matching, is shown in Figure 4a. (002) and (004) Bragg diffraction peaks were examined, as they are most sensitive to the K^+ extraction / insertion process. Note that in Figure 4b, peaks aside from (00 l) are difficult to analyse, owing to their low diffraction intensity. Amongst the (00 l) Bragg peaks, shifting of the (004) peak is the most noticeable upon K^+ extraction (charging) and reinsertion (discharging), as observed in Figure 4c. Upon charging, the (002) and (004) peaks shift to lower angles whilst the P2-type structure is retained, as observed in Figures 4b and 4c. Moreover, different degrees of shifting and asymmetric peak evolutions are apparent, as has also been observed in P2-type $K_{0.6}CoO_2$ ¹ and $K_{0.5}MnO_2$ ², providing further evidence of a multitude of phase transitions in $K_{2-x}Ni_2TeO_6$ as K ions are extracted. At some juncture during charging, significant peak broadening occurs after which a new set of XRD peaks to the left of the (002) and (004) peaks appear, signature of the phase transitions. Owing to the limitation posed by the low resolution and significant broadening of peaks, we desist to explicitly determine the structures of the new phases hallmarking the phase transitions. The phase evolution is reversed during the discharge process, suggesting reversible K extraction and insertion behaviour in P2- $K_2Ni_2TeO_6$. It is noteworthy that during the charge/discharge process, Bragg peaks tend to shift to lower/higher angles, indicating an underlying solid solution (monophasic or single-phase) mechanism. The appearance of new peaks (as is evident in Figure 4c) suggests a two-phase (biphasic) reaction in the composition range, but these peaks also shift continuously which is typical of a solid solution (monophasic) reaction. The overall trend is a continuous increase and decrease of the c lattice parameters upon charge and discharge, respectively. Upon charging, (00 l) Bragg diffraction peaks monotonically shift towards low angles, in principle due to the enhancement in the electrostatic repulsion between the Ni/TeO₆ octahedral slabs along the c -axis and *vice-versa* during discharge. Further, a decrease / increase in the a -axis concomitantly proceeds with charging / discharging (Supplementary Table 11),

due to the decrease / increase in nickel ion radius upon oxidation / reduction. The XRD results, in summary, demonstrate that P2-type $\text{K}_2\text{Ni}_2\text{TeO}_6$ can store K ions via a highly reversible topotactic reaction, which entails not only a monophasic but also a biphasic reaction. In addition, *ex situ* XRD results (Supplementary Figure 19) confirm that repeated K^+ extraction / insertion does not significantly affect the structural integrity of the P2-type $\text{K}_2\text{Ni}_2\text{TeO}_6$ cathode material.”

Reference

1. Kim, H. *et al.* K-ion batteries based on a P2-type $\text{K}_{0.6}\text{CoO}_2$ cathode. *Adv. Energy Mater.* **1700098** (2017).
2. Kim, H. *et al.* Investigation of potassium storage in layered P3-type $\text{K}_{0.5}\text{MnO}_2$ cathode. *Adv. Mater.* **1702480** (2017).

COMMENTS FROM REFEREE #2:

There has been a wave of interest recently in "honeycomb" variants of the layered oxide structures that dominate Li-ion and Na-ion battery cathode applications. The paper under review expands the scope of that interest to K-ion batteries, another growth area in solid-state electrochemical research. While it is arguable whether K-ion batteries have genuine commercial potential, given the obvious challenges such as slow kinetics and extreme hydrophilicity, I have no doubt that this work will attract attention from the huge number of battery researchers looking for new avenues to explore.

While the compounds described are certainly interesting, some of the claims about their properties are hyperbolic, or at least, framed very optimistically. High voltages and the resulting need for ionic liquid electrolytes entails as many disadvantages as advantages. It is difficult to imagine $K_2Mg_2TeO_6$ really being used as a solid electrolyte, given the difficulties in making solid electrolytes commercially viable in the more favourable Li and Na cases. One particular claim that struck me was that "their high densities ... improves the volumetric energy density" - which is misleading because the high density is due to the presence of a heavy metal (Te), not to a more closely-packed structure. The performance of these materials should be described realistically - they are undeniably good K conductors, their thermal and cycling stabilities are excellent, they could theoretically perform better, and (perhaps most importantly) they open up new chemistry to exploration.

Technically, the work is very thorough and well done. No additional experiments are needed to convince me that the key new compounds synthesised and tested ($K_2Ni_2TeO_6$ and $K_2Mg_2TeO_6$) have been correctly described in terms of their chemical, crystal and electronic structures. Additional compounds are described in sufficient detail in some cases (e.g., the Ni/Mg/Co solid-solutions), but obliquely in others. For example, the Discussion ends with a mention of other "plausible candidates" including $K_3M_2AO_6$ and $K_2M_2AO_6$, without explicitly suggesting what A might be. The obvious A cations to try are Sb and Bi, based on the known Na (for K) analogues; and, indeed, the Synthesis section mentions that $K_3M_2BiO_6$ and

$K_3M_2SbO_6$ were prepared. However, the former is not mentioned anywhere else in the paper (as far as I can tell) and the latter (as $K_3Ni_2SbO_6$) only appears in the SI, with no useful additional commentary. This suggests a desire to hold a significant amount of work back for later publication, while still laying claim to the first synthesis here. While I recognise that the paper is already quite long, I feel that some more transparency concerning these additional compounds would be desirable.

In summary, the work is interesting and of excellent quality. The technical content is publishable in its present form in a good journal. Its suitability for an "all of science" journal is somewhat marginal, and this is reflected in the need to describe the work in a very optimistic way, rather than let the results speak for themselves. I would not be upset to see this published in *Nature Communications*, but err on the side of it being more suitable for a slightly more specialised (top-tier) materials or energy focused journal.

We are greatly encouraged by the reviewer's overall assessment of our manuscript and for the specific comments to improve our paper. We have modified the discussion in the manuscript to clarify the points based on the reviewer's suggestions. The detailed response to each point and the related modification in the revised manuscript are as follows.

Concerning the density of the tellurates:

Response: The reviewer is right in that the contention that "their (i.e., the tellurates) high densities ... improves the volumetric energy density" appears hyperbolic; thus, misleading to readers. This sentence has therefore been deleted from the manuscript. The reviewer's concern aroused our curiosity to investigate more on the influence of Te on the density of potassium-based compounds. Based on the unit-cell volume, the molecular weight and the number of motifs per unit cell of the tellurates, we were able to deduce material densities of close to 5 g cm^{-3} . It is true that the high tap density of these potassium-based tellurates is contingent on the presence of a heavy metal chalcogen (Te), and not to a more closely-packed structure. Indeed, we note, for instance that the

Na analogue $\text{Na}_2\text{Co}_2\text{TeO}_6$ ($\text{Na}_{2/3}\text{Co}_{2/3}\text{Te}_{1/3}\text{O}_2$) shows a higher density than the isostructural Te-free $\text{Na}_{2/3}\text{CoO}_2$ (4.7378 g cm^{-3} versus 4.6595 g cm^{-3} ; a 1.68 % increase). We also note, *vide infra*, that KTeOF_3 possesses two polymorphs with crystallographic densities of 3.19 and 3.72 g cm^{-3} (a 16.6 % increase), demonstrating that the crystal structure adopted by the compounds also do have a significant influence on the tap density *stricto sensu*. The same goes with KVOPO_4 and KVOTeO_4 as well as KVOF_3 polymorphs (3.055 and 3.009 g cm^{-3}).

Whilst it is true that Te *per se* increases the density, the crystal structure adopted by the tellurate compounds influences to a greater extent. The figure below (Figure 1) shows the crystallographic densities attained in various potassium-based compounds screened.

Figure 1. Material density obtained from P2-type honeycomb layered tellurates in comparison to a smorgasbord of potassium-based compounds screened in this study. Further details can be found in the Excel file attached for the editors' reference.

Besides the attained high voltages, great cyclability and thermal stability, the tellurates stand out, due to their closely packed adopted structure. However, cognisant that the presence of heavy metals in compounds and the attained tap densities may be a polemical issue for readers, we find it prudent eliding this sentence from the manuscript.

Necessity for High Voltage:

Response: High voltage is indeed required to increase the energy density of battery systems. A high voltage configuration not only increases the energy density, but also reduces the number of cells per battery pack. As a matter of fact, a series connection of cells is commonly the norm to increase the voltage of large battery packs, for instance, those utilised for electric vehicular applications. High voltage battery configuration results in a reduction in the number of cells, thereby curtailing the volume and cost of the battery pack. For this reason, high voltage battery configurations will indeed have considerable advantage in energy storage devices. This point has been underscored in the DISCUSSION part of the manuscript (page 19, line 664-page 20, line 672) as follows:

“[...] A high voltage configuration not only increases the energy density, but also reduces the number of cells per battery pack. As a matter of fact, a series connection of cells to achieve the required voltage is commonly the norm to increase the voltage of large battery packs, for instance, those utilised for electric vehicular applications. High voltage battery configuration results in a reduction in the number of cells, thereby curtailing the volume and cost of the battery pack. For this reason, high voltage potassium-ion battery configurations will indeed have considerable advantage in energy storage devices.”

Particularly for potassium ion battery, high voltage is anticipated considering the lowest redox potential of K^+/K in non-aqueous battery systems¹. Developing a high voltage potassium-ion battery calls for the development of high voltage cathode materials which are scarce. However, as the reviewer mentions, high voltage has the disadvantage that the stability of the conventional organic electrolytes is compromised.

To circumvent the decomposition of the electrolyte and also spurious side reactions that tend to occur with potassium-based compounds particularly at high voltage regimes, a need therefore exists to utilise stable electrolytes that can be coupled with these high voltage tellurates. This work proposes ionic liquids based on potassium (bis)trifluorosulfonylimide as a candidate electrolyte for high voltage potassium battery. We are aware that the ionic liquids employed show high viscosity; thereby limiting the rate performance. This is an issue worth addressing, and more precisely we are investigating on the influence of various additives in lowering the viscosity of the potassium-based ionic liquids. We have highlighted this point in the METHODS section (page 24, line 832-line 841) as follows:

“[...] Owing to the pronounced decomposition of the common electrolytes like 0.8 mol dm^{-3} KPF_6 (Potassium hexafluorophosphate) (Sigma Aldrich, 99.99%) in ethylene carbonate (EC) / propylene carbonate (PC) (1:1 volume %), EC / diethylcarbonate (DEC) (1:1 volume %) or 1 mol dm^{-3} KFSI (potassium bis(fluorosulfonyl)imide) in EC / DMC (1:1) that we experimented, we have used a 0.5 mol dm^{-3} potassium bis(trifluoromethanesulfonyl)imide (hereafter denoted as KTFSI) in 1-methyl-1-propylpyrrolidinium bis(trifluoromethanesulfonyl)imide (hereafter $\text{Pyr}_{13}\text{TFSI}$) (Kanto Chemicals (Japan), 99.9%, <20 ppm H_2O) ionic liquid, which was more stable against decomposition at high voltages and compatible with most potassium-based cathode materials.”

Reference

1. Komaba, S. *et al.* Potassium intercalation into graphite to realize high-voltage/high-power potassium-ion batteries and potassium-ion capacitors. *Electrochem. Commun.* **60**, 172-175 (2015)

Of ionic liquids and potassium-based compounds:

Response: We concur with the reviewer that the utilisation of ionic liquid electrolytes entails as many demerits as merits. Ionic liquids, in principle, tend to show high viscosity (thereby thwarting the rate performance) and are relatively expensive. Electrolytes

applicable to potassium ion battery are typically composed of KPF_6 , KClO_4 , KBF_4 or KFSI salts dissolved in organic solvents. However, according to our knowledge the **conventional organic electrolytes based on these salts (for instance, KFSI and KPF_6) tend to be unstable when coupled with most potassium-based cathode materials.** This is part of the reason as to why the evaluation of cathode materials for potassium-ion battery is lagging behind. As we have noted, most of these potassium-based are hygroscopic and thus control of the water content in organic electrolytes is paramount to assessing their innate electrochemical performance. The painstaking five-year work to develop not only *de novo* potassium-based compounds but also compatible electrolytes, has led us to underpin the necessity of ionic liquids for potassium-ion battery. We therefore believe that the use of ionic liquids, a potential candidate of which we propose in this work, is a breakthrough that will spearhead further evaluation of high voltage cathode materials for researchers delving in this new and promising field of rechargeable potassium-ion batteries.

Transparency concerning $\text{K}_3\text{M}_2\text{BiO}_6$ and $\text{K}_3\text{M}_2\text{SbO}_6$ additional compounds:

Response: We accede to the reviewer that transparency is required for these compounds that we mention in the Methods section. **Our attempts to prepare phase pure samples of $\text{K}_3\text{M}_2\text{BiO}_6$ and $\text{K}_3\text{M}_2\text{SbO}_6$ were in vain.** We believe that optimisation of the synthetic approach employed might yield highly pure samples, information of which is provided in the Methods section. It should be noted that **fairly pure samples of the Ni analogues could be obtained**, the reason as to why we provide the XRD pattern in the Supplementary Information. We agree that it will be worthy to mention the synthesis protocols of $\text{K}_3\text{M}_2\text{BiO}_6$ and $\text{K}_3\text{M}_2\text{SbO}_6$, which will guide the readers on the synthesis route to possibly follow in preparing phase-pure samples.

We have **rephrased the METHODS section** (page 21, line 717-line 722) to read as follows:

“[...]...Synthesis of other conceivable compositions such as $\text{K}_3\text{M}_2\text{BiO}_6$ and $\text{K}_3\text{M}_2\text{SbO}_6$ were attempted under the same firing conditions by mixing MgO, CuO or ZnO and Sb_2O_5 (Aldrich, 99.999%) or Bi_2O_3 (Aldrich, 99.99%) with the previous precursors in the

stoichiometric proportions. However, only the Ni analogue of $K_3M_2SbO_6$ was formed as a fairly pure phase, whereas the other intermediate compounds resulted in two phase products made up of the precursor compounds.”

Realisation of potassium-ion conductors as viable solid electrolytes for potassium-ion battery:

Response: It is indeed difficult to envisage the utilisation of solid electrolytes, at this nascent stage of potassium-ion battery development. However, taking into account the high reactivity and the possible dendrite formation in using potassium metal as well as carbon-based anode materials, the development of solid electrolytes certainly will be critical to enhance the commercial deployment of safe potassium-ion batteries. We certainly do not know what the future holds for this new battery system. Nevertheless, we state with conviction that once there is a surge of researchers looking for new avenues to explore in the potassium-ion battery field (given that this potentially exciting and new energy storage technology can combine high energy density, cycle life, and good power capability, all while using abundant potassium resources), the development of safe solid electrolytes will certainly be the areas to explore. We are therefore convinced the development of potassium-ion conductors is pivotal to attaining this goal.

In response to the reviewer’s comments, we have rephrased the following sentence in the DISCUSSION section (page 10, line 332) as follows:

“[...].... realisation of potassium-ion conductors as viable solid electrolytes for potassium-ion battery is still far from reality. However, taking a cue from the high reactivity and the possible dendrite formation in using alkali metal as well as carbon-based anode materials, the development of solid electrolytes certainly will be quintessential to the commercial deployment of safe potassium-ion batteries.”

The Discussion ends with a mention of other "plausible candidates" including $K_3M_2AO_6$ and $K_2M_2AO_6$, without explicitly suggesting what A might be.:

Response: Thanks to the reviewer for the suggestions. To address the reviewer comments, we have appended the sentence in the DISCUSSION section (page 20, line 684) to read as follows [quote verbatim]:

“...The obvious *A* cations to try are Sb and Bi, based on the known Na (for K) analogues]...”

COMMENTS FROM REFEREE #3:

This manuscript reports the synthesis and performance tests of a group of tellurate compounds toward their possible application in K-ion batteries. K-ion batteries is an interesting topic and good cathode materials may attract attention of the field. The experimental works are in general quite solid, despite a few minor flaws.

We sincerely thank the reviewer for providing constructive remarks and valuable comments, which were of great help in revising the manuscript. Accordingly, the revised manuscript has been systematically improved with lucidity of discussion.

Major comments:

Two overarching concerns need to be addressed, besides detailed technical questions listed below.

1) The main focus of this paper seems to be blur with both the ionic conductor material and cathode material presented. It would be better if only one functionality/application is presented with deep discussion than two materials being presented with general discussion. The ionic conductor/cathode can be separated as two papers. Also, it would be totally worth it to publish the information in the supplementary information as papers in the future. These are valuable information but nowadays few people get to read the supplementary info.

Response: Thanks to the reviewer's comments we have realised that the former Introduction section needs to be improved by placing concrete emphasis on one or two functionalities, rather than a general discussion on many applications. We have **substantially changed the RESULTS section** to solve these shortcomings. More specifically, emphasis has now been placed on the potential of the tellurates as high voltage (thus, high energy density) cathode materials for the emerging potassium-ion battery. Furthermore, potassium ion conducting materials (especially those with high ionic conductivity) are quite promising for battery applications, the reason as to why details on the ionic conductivity of the related Mg tellurate analogue have also been

mentioned where appropriate. We have deleted texts mentioning on the optical or magnetic properties, which can be part of future work. Other details that are worth referring by the readers, have been furnished in the supplementary information as suggested.

The specific changes we have made are as follows:

- a. The Introduction section now highlights on the advantages of developing rechargeable batteries relying on potassium-ion as charge carriers. The challenges to overcome, particularly on the cathode side, are also underscored. We have revised Figure 1 by merging it with Figure 2, thus placing more emphasis on the crystal and K^+ diffusional pathways of $K_2Ni_2TeO_6$ which is rudimental to assessing not only its feasibility as a cathode material for potassium-ion battery, but also as a (potential) potassium-ion conductor. Ionic conductivity has an influence on the electrochemical performance of materials.
- b. The RESULTS section has been appended. The organisation of the subheadings has been rearranged (taking also into account Reviewers 1 and 2 suggestions). We highlight the electrochemical properties of the cathode materials by revising Figures 1 and 3.
- c. The ionic conductivity of the tellurates described has been highlighted using the true ionically conducting $K_2Mg_2TeO_6$, which is merely an extension of the work of the Ni tellurates. The ionic conductivity has thus been mentioned in light of the structural aspects (including the potassium-ion dynamics) of the prototype materials, whilst providing the relevant supporting details in the Supplementary Information (as the reviewer suggests). The essence of using the honeycomb layered tellurates is the multifunctionality endowed in these materials, an aspect that we explain in the Introduction section for clarity to the readers.
- d. As the reviewer suggests, adding more details regarding the multifunctionality may adversely affect the readability of this work, hence our future work will entail the further exploration of various multifunctionalities such as magnetism, optics, superconductivity, etc., insights of which we can report as another paper.

2) Tellurium is a very expensive element, which implies the tellurates may never be really commercialized for K-ion batteries. The reviewer totally understands fundamental and exploration type of research should not be limited by application considerations. However, more discussion on the general impact of this work in K-ion battery and energy storage is necessary for a broad readership.

Response: We thank the reviewer for giving us a chance to clarify this point. We indeed recognise that tellurium is not a constituent element of choice for potassium-ion batteries because of the high tellurium price. Thus, it suffices to say that the present work is more of a **fundamental study**. This point has now been underscored in the DISCUSSION section (page 19, line 652-line 654) as follows:

“...[...] the use of non-terrigenous element such as Te, this study represents more of a fundamental interest....”

Currently, there is an **uptick in demand for tellurium** driven by new applications that use high-purity Te. We note that tellurium (a chalcogen) has several important commercial uses. It is primarily used for manufacturing films essential to photovoltaic solar cells as well as thermoelectric applications. For instance, when alloyed with cadmium—the leading end-use among these applications cadmium telluride-based solar cells can be produced¹. Tellurium is further used in copying machines and as a colouring agent in ceramics and glass, and as a vulcanising agent in the chemical industry to make durable products, including an additive that improves rubber’s heat resistance². Moreover, medical instrumentation, integrated circuits, and laser diodes, all of which have experienced robust manufacturing growth in recent years, contain tellurium^{3,4}. In addition, lead and germanium telluride-based materials, which display intriguing functionalities, have also been intensively studied from both fundamental and technological perspectives^{5,6}. Although **tellurium may not be a terrigenous element**, it has intensively been studied from both **fundamental and technological perspectives**, owing to the intriguing functionalities endowed in tellurium-based compounds

On a **materials cost perspective**, the content of tellurium used in the tellurates is relatively low to significantly influence the total material cost of preparation. Find attached an Excel file with the material cost calculations for some of the tellurates detailed in this work. Indeed, 1 g of $K_2Ni_2TeO_6$ costs around ¥49 (translating to \$0.43 or thereabout) which is a more than 50% reductions in the material costs, in comparison to lithium battery cathode materials such as the stellar $LiCoO_2$ or $LiNi_{0.8}Co_{0.1}Co_{0.1}O_2$ (~¥110 (~\$1)/g). Thus, the presence of tellurium is not a deterrent to the total material cost. Furthermore, the packing casings of batteries are the main components that significantly influence the cost/price of a battery pack. The use of aluminium as a current collector on the anode side for K-ion battery is an advantage, particularly of largescale battery packs where a significant cost reduction can be expected⁵.

This study has shown that **reversible potassium ion (de)insertion can be attained at high voltages in honeycomb layered compounds such as the tellurates**, which is a major leap in potassium-ion battery where there is a dearth of high energy density cathode materials. This is apparent in Figure 1 below which univocally shows the tellurates reigning supreme as high voltage cathode materials from a plethora of potassium-based compounds we screened. We reiterate that, from a fundamental standpoint, this study stresses that the **design of high voltage potassium-based layered compounds**, demonstrating reversible K^+ (de)insertion, **is not trivial in the absence of electronegative moieties such as TeO_6^{6-}** and augmented with theoretical support. These new P2-type layered compounds, adopting the honeycomb structure, provide valuable information in the hunt for even better cathode materials for rechargeable potassium-ion battery.

Figure 1. Average voltages attained by P2-type honeycomb layered tellurates in comparison to other potassium-based cathode materials. Further details can be found in the Excel file attached for the editors' reference.

The fundamental understanding gained from this work can also be utilised to further design new high performance cathode materials for rechargeable potassium-ion batteries. We have highlighted this point in the DISCUSSION section (page 19, line 652-line 660) as follows:

“With a theoretical capacity in the range of 120~150 mA h g⁻¹ and the use of non-terrogenous element such as Te, this study represents more of a fundamental interest. Nevertheless, related materials can be made useful for practical applications provided a less heavy congener element can be found to replace Te⁶⁺ (*d* shell occupancy). From a fundamental standpoint, however, this study stresses that the design of potassium-based cathode materials, demonstrating reversible K⁺ (de)insertion at preponderant high voltages, is not trivial in the absence of electronegative moieties

such as TeO_6^{6-} and augmented with theoretical support. The low theoretical capacities displayed by the tellurates are compensated by their high cell voltages (3.6~4.3 V versus K^+/K). These new P2-type layered compounds, adopting the honeycomb structure, provide valuable information in the search for even better cathode materials for rechargeable potassium-ion battery.”

Reference

1. Basol, B. M. *et al.* Brief review of cadmium telluride-based photovoltaic technologies. *J. of Photonics for Energy*, **4(1)**, 040996 (2014).
2. Akiba, M. *et al.* Vulcanization and crosslinking in elastomers. *Prog. Polym. Sci.* **22**, 475521 (1997).
3. Zweibel, K. The Impact of Tellurium Supply on Cadmium Telluride Photovoltaics. *Science* **328(5979)**, 699-701 (2010).
4. Snyder, G. J. *et int.* Lead telluride alloy thermoelectrics. *Materialstoday* **14(11)**, 526-532 (2011).
5. Joshi, G. *et al.* Enhanced Thermoelectric Figure-of-Merit in Nanostructured p-type Silicon Germanium Bulk Alloys. *Nano Lett.* **8(12)**, 4670-4674 (2008).
6. Passerini, S. *et int.* A cost and resource analysis of sodium-ion batteries. *Nat. Rev. Mater.* **3**, 18013 (2018).

Other comments:

Detailed comments/questions are as follows:

- 1) The reviewer strongly suggest the authors to follow the guide for authors to write the manuscript. For example, this current manuscript does not have page number, making the reviewing painful.

Response: We apologise for our failure to number the pages. We have added the page numbers and also adjusted the layout of the manuscript and the supplementary information to comply with the standards of Nature Communications.

2) Page 3. Typo: $K_{2/3}M_{1/3}Te_{1/3}O_2$ should be $K_{2/3}M_{2/3}Te_{1/3}O_2$.

Response: We thank the reviewer's careful examination on our manuscripts. We have corrected the typographical error (page 4, line 115) and gone through the manuscript carefully for any related errors.

3) A completely in-plane ordering of Ni and Te was used in the refinement with putting Ni and Te at different sites. Did the author check with refinements that puts Ni and Te in random distribution in $P6_3/mcm$ space group to see if it is possible? The ionic radius of Ni^{2+} is 0.69 Å and Te^{6+} is 0.56 Å in octahedral sites. I guess they would prefer to order from both size and charge coupling considerations, but some degree of disordering still may exist, which could give significant impact to the diffusion barriers. The disordering of K^+ at K1, K2 and K3 sites seems also indicating the disordering in the Ni-Te plane. Local structure sensitive characterizations such as Pair Distribution Function or solid state NMR may be considered to complement XRD refinement.

Response: To assuage the reviewer's concern regarding Ni / Te in-plane disorder, disorder arising from cationic mixing between Te and Ni in the octahedral site slabs was checked again. When a disordered Ni / Te slab random model was assumed, the refinement did not converge to any reasonable agreement factors. The best fit could only be obtained when an ordered honeycomb structure was considered, yielding agreement indices of $\chi^2 = 1.77$ and $R_B = 4.29\%$ (as is shown in Figure 1). Moreover, refining the occupancy factor (g) at Te site for the ordered $K_2Ni_2TeO_6$ honeycomb structure resulted in no improvement in the agreement factors. It is thus reasonable to

envison an in-plane ordering of (Ni/Te)O₆ layers akin to isotopic ordered Na₂Ni₂TeO₆, that possesses Ni²⁺ and Te⁶⁺ in an octahedral (6-fold) oxygen coordination. K₂Ni₂TeO₆ has two possible arrangements of charges under the extreme ionic model as K¹⁺₂Ni²⁺₂Te⁶⁺O²⁻₆ or K¹⁺₂Ni³⁺₂Te⁴⁺O²⁻₆. However, in the present work, this compound exists as K¹⁺₂Ni²⁺₂Te⁶⁺O²⁻₆ based on spectroscopic characterisations, whilst the stoichiometric composition was confirmed to be in line with the expected composition by using inductively coupled plasma (ICP) analysis.

Figure 1. Rietveld refinement of the synchrotron XRD pattern of K₂Ni₂TeO₆ indexed in the ordered honeycomb framework adopting the *P6₃/mcm* hexagonal space group. Inset shows the electron diffraction patterns recorded on K₂Ni₂TeO₆ indexed in a hexagonal lattice along [100] and [001] directions. These results have been added to RESULTS section and Supporting Information section of our revised manuscript (manuscript page 6-Figure 1, Supplementary Figure 2 and Supplementary Table 2).

A Ni/Te in-plane ordering in this compound is not unprecedented, considering the ionic radius of Ni²⁺ is 0.69 Å and Te⁶⁺ is 0.56 Å in octahedral sites. Unlike the honeycomb layered oxides comprising Ni²⁺/Sb⁵⁺O₆ octahedral slabs where disorder in the slabs

should be expected (the closer Shannon ionic radii of Ni^{2+} and Sb^{5+} in a 6-fold coordination), **stoichiometric honeycomb layered oxides with slabs (as well as other frameworks) incorporating the hexavalent Te^{6+} tend to show no disorder** within the transition metal slabs. Typical examples include, but are not limited to, the following: $\text{Na}_2\text{Ni}_2\text{TeO}_6$ ¹, $\text{Na}_2\text{Mg}_2\text{TeO}_6$ ², $\text{Na}_2\text{Co}_2\text{TeO}_6$ ³, $\text{Ba}_2\text{NiTeO}_6$ ⁴, $\text{Ba}_2\text{MnTeO}_6$ ⁵, $\text{Ba}_2\text{CoTeO}_6$ ⁶, $\text{Ba}_2\text{ZnTeO}_6$ ⁷, $\text{Ba}_2\text{CuTeO}_6$ ⁸, $\text{Pb}_2\text{MnTeO}_6$ ⁹, $\text{Pb}_2\text{CoTeO}_6$ ¹⁰, $\text{Sr}_2\text{CuTeO}_6$ ¹¹, $\text{Sr}_2\text{CoTeO}_6$ ¹², $\text{Sr}_2\text{MgTeO}_6$ ¹³, Ni_3TeO_6 ¹⁴ and $\text{Sr}_2\text{NiTeO}_6$ ¹⁵.

Whilst the local Ni/Te disorder was beyond the detectable limits of the X-ray diffraction analysis, further attempts were made to study the nature of the Ni/Te local order through **electron diffraction (high resolution TEM (transmission electron microscopy) imaging)** and ^{125}Te solid-state NMR. No evidence of stacking disorder could be detectable through TEM studies. This is most clearly seen in the electron diffraction patterns for the hexagonal [100] zone axis, which reveal neither streaks nor rods of scattering along the *c*-axis direction that are expected when stacking faults are present. Additionally, no evidence of superstructures could be detectable from the **synchrotron XRD refinements** and HRTEM, unequivocally showing the hexagonal **$P6_3/mcm$ space group** to be the most appropriate structure for the present **ordered $\text{K}_2\text{Ni}_2\text{TeO}_6$** . This is indeed consistent with prior single-crystal and powder diffraction investigations of the isotypical $\text{Na}_2\text{Ni}_2\text{TeO}_6$ ordered phase¹⁶.

We further deliberated on solid state ^{125}Te NMR measurements to probe the local structure around Te atoms. Our attempts were however elusive, owing to the strong paramagnetic interaction related to the presence of Ni^{2+} , rendering **solid-state ^{125}Te NMR** not suitable to probe the local structure for $\text{K}_2\text{Ni}_2\text{TeO}_6$. The present hurdle has highlighted the importance of studying the local structures in analogous tellurate compounds containing no paramagnetic elements by means of solid-state NMR, which we intend to address as part of future work.

Nevertheless, based on the results garnered from the various characterisation techniques employed in this study so far, it is our firm conviction that the as-synthesised $\text{K}_2\text{Ni}_2\text{TeO}_6$ adopts an ordered honeycomb lattice.

References

1. Karna, S. K. *et al.* Sodium layer chiral distribution and spin structure of $\text{Na}_2\text{Ni}_2\text{TeO}_6$ with a Ni honeycomb lattice. *Phys. Rev. B* **95**, 104408 (2017).
2. Li, Y. *et al.* New P2-Type Honeycomb-Layered Sodium-Ion Conductor: $\text{Na}_2\text{Mg}_2\text{TeO}_6$. *ACS Appl. Mater. Interfaces* **10**, 15760–15766 (2018).
3. Lefrancois, E. *et al.* Magnetic properties of the honeycomb oxide $\text{Na}_2\text{Co}_2\text{TeO}_6$. *Phys. Rev. B* **94**, 214416 (2016).
4. Asai, S. *et al.* Magnetic ordering of the buckled honeycomb lattice antiferromagnet $\text{Ba}_2\text{NiTeO}_6$. *Phys. Rev. B* **93**, 024412 (2016).
5. Wulff, L. *et al.* On the Crystal Chemistry of Tellurates Containing Mn^{2+} in the Cationic and Anionic Part of the Crystal Structure: $(\text{Mn}_{2.4}\text{Cu}_{0.6})\text{TeO}_6$, $\text{Ba}_2\text{MnTeO}_6$ and $\text{Pb}(\text{Mn}_{0.5}\text{Te}_{0.5})\text{O}_3$. *Zeitschrift für Naturforschung B* **53(1)**, 49–52 (2018).
6. Chanlert, P. *et al.* Field-driven successive phase transitions in the quasi-two-dimensional frustrated antiferromagnet $\text{Ba}_2\text{CoTeO}_6$ and highly degenerate classical ground states. *Phys. Rev. B* **93**, 094420 (2016).
7. Von Rother, H.-J. *et al.* Das Schwingungsspektrum von $\text{Ba}_2\text{ZnTeO}_6$. *Z. Anorg. Allg. Chem.* **436**, 213 (1977).
8. Gibbs, A. S. *et al.* $S = 1/2$ quantum critical spin ladders produced by orbital ordering in $\text{Ba}_2\text{CuTeO}_6$. *Phys. Rev. B* **95**, 104428 (2017).
9. Retuerto, M. *et al.* $\text{Pb}_2\text{MnTeO}_6$ Double Perovskite: An Antipolar Anti-ferromagnet. *Inorg. Chem.* **55**, 4320–4329 (2016).
10. Wedel, B. *et al.* On the Crystal Structures of the Tellurates $\text{Pb}_3\text{Fe}_2\text{Te}_2\text{O}_{12}$ and $\text{Pb}_2\text{CoTeO}_6$. *Zeitschrift für Naturforschung B*, **52(1)**, 35–39 (1997).
11. Koga, T. *et al.* Magnetic structure of the $S = 1/2$ quasi-two-dimensional square-lattice Heisenberg antiferromagnet $\text{Sr}_2\text{CuTeO}_6$. *Phys. Rev. B* **93**, 054426 (2016).
12. Augsburger, M. S. *et al.* Preparation, crystal and magnetic structures of two new double perovskites: $\text{Ca}_2\text{CoTeO}_6$ and $\text{Sr}_2\text{CoTeO}_6$. *J. Mater. Chem.* **15**, 993-1001 (2005).
13. Ubic, R. *et al.* Re-Examination of the Structure of $\text{Sr}_2(\text{MgTe})\text{O}_6$ Double Perovskite. *J. Aus. Ceram. Soc.* **47(2)**, 49-56 (2011).
14. Živkovi, I. *et al.* Ni_3TeO_6 —a collinear antiferromagnet with ferromagnetic honeycomb planes. *J. Phys.: Condens. Matter* **22**, 056002 (2010).

15. Ortega-San Martin, L. *et al.* Crystal Structure of the Ordered Double Perovskite, Sr₂NiTeO₆. *Z. Anorg. Allg. Chem.* **631(11)**, 2127-2130 (2005).

16. Sankar, R. *et al.* Crystal growth and magnetic ordering of Na₂Ni₂TeO₆ with honeycomb layers and Na₂Cu₂TeO₆ with Cu spin dimers. *CrystEngComm.* **16**, 10791-10796 (2014).

4) In addition, it would be great if the authors can provide the cif files of the crystal structure after the refinements, as part of the supplementary information.

Response: We agree with the reviewer that it would have been great to be able to provide the relevant crystallographic information files (cif) to aid other researchers visualise the crystal structures as well as advance the current study. We have therefore **provided the cif files as part of the Supporting Information**. In addition, we have mentioned in the METHODS section the pertinent accession codes for the new structures/compositions presented.

5) Figure 1a is too small to read. A figure of refinement could be very informative but not with this small size. Suggest to either increase the size in main text, or give a high resolution figure in supplementary info.

Response: Thank you for the suggestion. We have **increased the size of Figure 1a**, so that the readers can have a clear glimpse of the quality of the refinement.

6) How hygroscopic are the tellurates? How long has the sample been exposed to air before the XRD in figure S11 was collected? Minutes, hours, or days?

Response: We appreciate your comments very much. The tellurates detailed in this study are hygroscopic. For instance, K₂Ni₂TeO₆ tends to amorphise after a **1-week (7**

days) exposure in humid air. To make it clear to the readers, we have appended the caption of **Supplementary Figure 9** to read as follows:

“[...] Conventional XRD patterns of pristine $K_2Ni_2TeO_6$, revealing a diminution in the intensity of diffraction peaks upon a 1-week moist air exposure. [...]”

7) The computed in-plane diffusive energy barrier is 0.35 eV, while the experimental measured activation energy is 0.92 eV. Although they are not exactly the same energies/barriers, they commonly do not deviate that much in other ionic conductors. Could the authors explain this?

Response: Thank you for raising this concern. The computed diffusive energy barriers are based on BVEL approach and these values do not have a real physical meaning. BVEL approach has been used as a legitimate tool to only gauge the ease of K^+ to diffuse within the tellurate framework (thereby visualising the K^+ diffusional trajectories) in comparison to some known potassium-based cathodes and potassium ion conductors we have listed in the Supplementary Table 6. In order for the readers to clearly grasp this point, we have added this sentence in the main manuscript as follows (page 8, line 250-line 255):

“[...]...It is important to accentuate here that although the diffusive barrier values computed using BVEL approach have unclear physical meaning, the values are deemed legitimate in assessing the cation mobility of materials.”

We reiterate that were it not for the BVEL approach, pinpointing the tellurates from a ‘haystack’ of potassium-based compounds would have indeed been a formidable fete. The BVEL values are further validated by complementary DFT-NEB (Nudged Elastic Band) calculations (see Supplementary Figure 12) that show similar diffusion barriers; thus the computed values based on the well-established BVEL approach are accurate.

We concur with the reviewer that there is reason for concern about the disparate values between experimental and theoretical diffusive barriers. The reasons behind this we list as below:

(1) The calculated values are based on a 100% compactness. We however were able to only prepare pellets with a compactness of 70%. The compactness (ceramic density) of pellets are known to significantly impact the diffusive barriers. For instance, Li *et al*¹ recently report the Na analogue (*viz.*, P2-type $\text{Na}_2\text{Mg}_2\text{TeO}_6$) to show an activation energy of 0.341 eV (for pellets with a relative density of 87.2%), whilst Evstigneeva and co-workers², *vide infra*, reported an activation energy >0.6 eV for ceramics with a relative density of 74%. This point we have stated in the manuscript as follows (page 10, line 329- line 331):

“[...]...further denser ceramics can, in principle, be expected to show even higher K-ion conductivities. Theoretical calculations indeed show that denser ceramics can show high K-ion conductivity (see **Supplementary Notes 2 and 3**, and **Supplementary Figures 14 and 15**).”

Preparation of high compactness (dense) pellets calls for a more stringent sintering techniques that we are still currently exploring.

(2) Quantifying the BVEL values, through performing ionic conductivity measurements for the true ionically conducting $\text{K}_2\text{Mg}_2\text{TeO}_6$, has shown that K^+ dynamics within these tellurates entail more complex phenomena. Indeed, complementary DFT-NEB (Nudged Elastic Band) calculations (furnished in the Supplementary Information (Supplementary Note 3 and Supplementary Figure 15)). The theoretical computations based on the assumption of a fully cooperative K^+ motion surmise low diffusion barriers even at high temperatures. To fully validate the low activation barriers in these tellurates, we attempted to conduct high temperature conductivity studies; however our efforts were in vain owing to the instrumentation limits coupled with the high inductance observed beyond 300°C . We are yet to design an apparatus that can allow to perform precise high temperature conductivity.

Reference

1. Li, Y. *et al.* New P2-Type Honeycomb-Layered Sodium-Ion Conductor: $\text{Na}_2\text{Mg}_2\text{TeO}_6$. *ACS Appl. Mater. Interface* **10**, 15760-15766 (2018).

2. Evstigneeva, M. A. *et al.* A New Family of Fast Sodium Ion Conductors: $\text{Na}_2\text{M}_2\text{TeO}_6$ (M = Ni, Co, Zn, Mg). *Chem. Mater.* **23**, 1174-1181 (2011).

8) How is the long term stability of the cathode material? Could the author provide an *ex situ* XRD of the material after some cycles?

Response: As recommended by the reviewer, we have provided *ex situ* XRD patterns of $\text{K}_2\text{Ni}_2\text{TeO}_6$ upon subsequent cycling. No change in the XRD powder pattern was found for samples that had been cycled for more than 20 and 30 cycles, which is consistent with a sustained reversibility of the system (as stated in the main manuscript (page 15, line 512-line 515) and shown in Supplementary Figure 19).

9) The EIS spectra of $\text{K}_2\text{Mg}_2\text{TeO}_6$ in 25°C and 100°C are expected to show semi-circles with larger radii than those of 200°C and 300°C. Why do they only show dots in very low impedance area?

Response: Thank you for identifying this area of potential ambiguity. EIS spectra of $\text{K}_2\text{Mg}_2\text{TeO}_6$ in 25 °C and 100 °C indeed show semi-circles with larger radii than those of 200 °C and 300 °C. We have changed the labeling position to avoid any confusion that may arise. See Supplementary Figure 11.

10) It is suggested that the spectra of the standards, such as NiO, should be also shown in the Ni XANES results (figure 6c) for reader to gauge the increase of the valence of Ni.

Response: In response to the reviewer's comment, we have availed the XANES spectra of the reference NiO with Ni^{2+} adopting the same octahedral oxygen coordination, as observed in $\text{K}_2\text{Ni}_2\text{TeO}_6$. We have also furnished additional high quality Ni K-edge

XANES spectra for clarity. As shown in Figure 2 below, the Ni K-edge XAS spectra (plotted with reference NiO) affirm the divalent state of nickel in $K_2Ni_2TeO_6$.

Figure 2. Normalised Ni K-edge XANES spectra recorded on $K_2Ni_2TeO_6$ upon (a) charging and (b) discharging. NiO spectra has been shown as a reference for Ni^{2+} adopting octahedral coordination.

These results have been added to RESULTS section of our revised manuscript (page 14, Figures 4d and 4e).

11) The space in Figure 6b is not well used to better demonstrate the details as 6a and 6b are not that different. It would be much better to show in 6b only zoom-in of two peaks with the change in their 2theta.

Response: Thanks to the reviewer's suggestions, we have provided a magnified image of the peaks that demonstrate changes that are pertinent to capturing the phase transition phenomena in $K_{2-x}Ni_2TeO_6$ during galvanostatic cycling. See Figure 3 below.

Figure 3. XRD patterns from 10° to 38° and enlarged images of (00 l) Bragg peaks obtained from $K_2Ni_2TeO_6$ upon charge and discharge.

These results have been added to RESULTS section of our revised manuscript (page 14, revised Figure 4a).

12) The authors claimed that the reversible capacity is partly due to O-redox based on the O-K-edge XANES. This is a weak argument. Since the O 2p and Ni 3d orbitals are hybrid, it is arguable to say this is an O-redox. Similar arguments have been extensively discussed in Li-excess cathode materials in Li-ion batteries.

Response: As the reviewer points out, O-redox in Li-excess cathode materials for Li-ion batteries has indeed been a subject of passionate research. Tarascon and co-workers revealed that the dominant factor triggering O-redox in such materials arises from the covalency existing between transition metal d orbital and oxygen p orbitals¹. For instance, in Li_2RuO_3 , the covalent bonding between Ru 4d and O 2p orbitals states leads to stabilisation of the oxygen holes and thus the redox reaction proceeds reversibly. As

for Li_2MnO_3 , *vide infra*, the covalent bond between Mn 3d and O 2p is weak; therefore $(\text{O}_2)^{\cdot-}$ is not stable at high voltage, consequently resulting to the release of oxygen. Ceder and co-workers have reported that the excess Li exists in the vicinity of O within the crystal structure of Li-excess cathode materials and the formation of a new energy state emanates from the linear structure of Li-O-Li². Bruce and co-workers further report that the interaction between Mn-O and Li-O has weaker covalent bonds than Ni-O and Co-O in $0.5\text{Li}_2\text{MnO}_3\text{-}0.5\text{LiNi}_{1/3}\text{Co}_{1/3}\text{Mn}_{1/3}\text{O}_2$; oxygen ligand holes are thereby localised around Mn or Li³. Charge compensation in this system is ascribed to the oxidation of Co and Ni at the initial charge process, whereas the **formation of oxygen holes occurs at high voltage**. O K-edge XANES results have also been reported in Li-excess cathode materials. For instance, Oishi and co-workers show that in Li_2MnO_3 , the increase of the intensity of the spectral features centered at 532 eV on the higher energy side beyond the pre-edge peak is mainly caused by O-redox⁴. Yabuuchi and co-workers later confirmed a similar peak increase emerging on the high energy side beyond the pre-edge peak at the end of charging of $\text{Li}_{1.2}\text{Nb}_{0.4}\text{Mn}_{0.4}\text{O}_2$ ⁵. In these cases, Mn is the main component of the transition metal. Compared with Mn-O which possesses a weak covalent bond, **Ni-O bond has strong covalency**, resulting in disparate behaviour. Uchimoto and co-workers reported an increase of peak intensity on the lower energy side than the pre-edge peak upon delithiation reaction of LiNiO_2 ⁶. This trend is similar to the change of O K-edge XANES observed in this study, affirming the **contribution of ligand hole by Ni-O forming a strong hybrid orbital**.

The reviewer is certainly right that the **O 2p and Ni 3d orbitals are hybridised**, and thus the changes reflected in the O K-edge XANES spectra cannot just be concluded to evince an O-redox process. Various features observed in O K-edge spectra (shown in Figure 4 below) can be assigned to the transition of electron from the O 1s core level to the unoccupied O 2p orbital states, arising from the hybridisation of nickel and with O 2p orbitals and are described in detail in references 6 and 7. Te^{6+} does not possess single d electrons as it exhibits closed-shell configuration ($[\text{Kr}] 4d^{10}$) and is thus unable to interact with O through its d orbitals. Orbital hybridisation only occurs between O and Ni.

Figure 4. Normalised O K-edge XANES spectra recorded on $\text{K}_2\text{Ni}_2\text{TeO}_6$ upon charging and discharging.

These results have been added to RESULTS section of our revised manuscript (Figure 5c).

A salient feature is the pre-edge peak observed around 528 eV upon charging (K^+ extraction), due to hybridisation of Ni 3d–O 2p orbitals and is assigned to $3d^8\bar{L} \rightarrow \underline{c}3d^8$ transition (\bar{L} denoting a ligand hole whilst \underline{c} denotes the core). This behaviour is in excellent concordance with that observed in Li-excess LiNiO_2^6 and SrNiO_3^8 . The increase in the intensity of the peak observed at 528 eV, in principle reflects the degree of hybridisation between Ni 3d – O 2p orbitals, hence we can state that this strength is strongest upon K^+ extraction from $\text{K}_{2-x}\text{Ni}_2\text{TeO}_6$. K^+ extraction leads to the increase in the valency state from $\text{Ni}^{2+} \rightarrow \text{Ni}^{3+}$. Thus, the ground state of Ni^{3+} will be predominantly $3d^7$. However, owing to configuration mixing caused by the hybridisation with O 2p orbitals, the ground state has a considerable $3d^8\bar{L}$ character (as has been observed in Li-excess compounds). Presence of this $3d^8\bar{L}$ configuration in the ground state causes this

pre-edge structure prominent at 528 eV upon charging. It is therefore assigned to the transition $3d^8 \underline{L} \rightarrow \underline{c} d^8$ and can be treated as a signature of the hybridisation strength of Ni $3d$ -O $2p$ orbitals⁶. In summary, the trend observed in the O K -edge spectra signifies that, hybridisation or covalency is large upon potassium extraction from $K_2Ni_2TeO_6$ and *vice versa* holds.

To address this concern we have appended the RESULTS section (page 18, line 605-line 622) to read as follows:

“...A clear change of the O K -edge XAS of the $K_2Ni_2TeO_6$ cathode material before and after charging and discharging was observed. The peak around 532 eV arises from the hybridisation of O $2p$ and Ni $3d$ orbitals, whilst the peak on the high energy side emanates from the hybridisation of O $2p$ orbital and Ni $4s,p$ orbitals⁶. A new peak is discernible at 528 eV (on the low energy side) upon charging, ascribable to the formation of oxygen ligand hole. This behaviour is in excellent concordance with that observed in Li-excess $LiNiO_2$ ⁶ and $SrNiO_3$ ⁸. The increase in the intensity of the peak observed at 528 eV, in principle reflects the degree of hybridisation between Ni $3d$ – O $2p$ orbitals, hence we can state that this strength is strongest upon K^+ extraction from $K_{2-x}Ni_2TeO_6$. K^+ extraction leads to the increase in the valency state from $Ni^{2+} \rightarrow Ni^{3+}$. Thus, the ground state of Ni^{3+} will be predominantly $3d^7$. However, owing to configuration mixing caused by the hybridisation with O $2p$ orbitals, the ground state has a considerable $3d^8 \underline{L}$ character (as has been observed in Li-excess compounds such as $LiNiO_2$ ⁶ upon charging). Presence of this $3d^8 \underline{L}$ configuration in the ground state invokes this pre-edge structure conspicuous at 528 eV upon charging. It is therefore assigned to the transition $3d^8 \underline{L} \rightarrow \underline{c} d^8$ and can be treated, *vide supra*, as a signature of the hybridisation strength of Ni $3d$ – O $2p$ orbitals⁶. In summary, the trend observed in the O K -edge spectra (as was also *a priori* revealed by first-principle calculations) signifies that, hybridisation or covalency is large upon potassium extraction from $K_2Ni_2TeO_6$ and *vice versa* holds.”

Reference

1. Tarascon, J-M. *et int.* The intriguing question of anionic redox in high-energy density cathodes for Li-ion batteries. *Energy Environ. Sci.* **9**, 984-991 (2016).

2. Ceder, G. *et al.* The structural and chemical origin of the oxygen redox activity in layered and cation-disordered Li-excess cathode materials. *Nat. Chem.* **8**, 692-697 (2016).
3. Bruce, P. *et al.* Charge-compensation in 3d-transition-metal-oxide intercalation cathodes through the generation of localized electron holes on oxygen. *Nat. Chem.* **8** 684-691 (2016).
4. Oishi, M. *et al.* Direct observation of reversible oxygen anion redox reaction in Li-rich manganese oxide, Li_2MnO_3 , studied by soft X-ray absorption spectroscopy. *J. Mater. Chem.* **4**, 9293-9302 (2016).
5. Yabuuchi, N. *et al.* Origin of stabilization and destabilization in solid-state redox reaction of oxide ions for lithium-ion batteries. *Nat. Comm.* **7**, 13814 (2016).
6. Uchimoto, Y. *et al.* Changes in electronic structure by Li ion deintercalation in LiNiO_2 from nickel *L*-edge and O *K*-edge XANES. *J. Power Sources* **97-98**, 326-327 (2001).
7. Kanda, H. *et al.* Cluster calculation of oxygen K-edge electron-energy-loss near-edge structure of NiO. *Phys. Rev. B* **58(15)**, 9693-9696 (1998).
8. Oriyasa, Y. *et al.*, X-ray Absorption Spectroscopic Study on $\text{La}_{0.6}\text{Sr}_{0.4}\text{CoO}_{3-\delta}$ Cathode Materials Related with Oxygen Vacancy Formation. *J. Phys. Chem. C* **115 (33)**, 16433–16438 (2011).

Hikari Sakaebe, Ph D

Chief Senior Researcher

Department of Energy and Environment

Research Institute of Electrochemical Energy

National Institute of Advanced Industrial Science and Technology (AIST)

Midorigaoka 1-8-31, Ikeda-shi, Osaka-fu, 563-8577, Japan

Phone: 81 72 751 9673

Fax: 81 72 751 9609

E-mail: hikari.sakaebe@aist.go.jp

REVIEWERS' COMMENTS:

Reviewer #1 (Remarks to the Author):

All the raised questions have been addressed properly, except the last in-situ XRD request, but the authors conducted the complementary information about the phase evolution during cycling from ex situ X-ray diffraction studies at a selection of various points collected at almost every 0.1 V voltage change during the charge and discharge processes, which is also beneficial for the understanding of the phase transition process.

Therefore, this manuscript can be accepted.

Reviewer #2 (Remarks to the Author):

I am satisfied with the authors' responses to the points I raised, as well as the points raised by the other Referees, and am comfortable seeing the work in its current form published in Nature Communications.

Reviewer #3 (Remarks to the Author):

The questions/comments raised in previous round of review have been nicely addressed by the authors and manuscript has been significantly improved.

The reviewer suggests this manuscript to be accepted for publication in Nature Communications.